# Development of a RIPK1 degrader to enhance antitumor immunity

Xin Yu[1,2,8], Dong Lu[1,8] ✉, Xiaoli Qi[1,2], Rishi Ram Paudel[1,2], Hanfeng Lin [1,2], Bryan L. Holloman[1,2], Feng Jin[1], Longyong Xu[3,4], Lang Ding [5], Weiyi Peng [6], Meng C. Wang [5], Xi Chen [3,4,7] & Jin Wang [1,2,3] ✉

The scaffolding function of receptor interacting protein kinase 1 (RIPK1) confers intrinsic and extrinsic resistance to immune checkpoint blockades (ICBs) and emerges as a promising target for improving cancer immunotherapies. To address the challenge posed by a poorly defined binding pocket within the intermediate domain of RIPK1, here we harness proteolysis targeting chimera (PROTAC) technology to develop a RIPK1 degrader, LD4172. LD4172 exhibits potent and selective RIPK1 degradation both in vitro and in vivo. Degradation of RIPK1 by LD4172 triggers immunogenic cell death, enhances tumor-infiltrating lymphocyte responses, and sensitizes tumors to anti-PD1 therapy in female C57BL/6J mice. This work reports a RIPK1 degrader that serves as a chemical probe for investigating the scaffolding functions of RIPK1 and as a potential therapeutic agent to enhance tumor responses to ICBs therapy.

Immune checkpoint blockades (ICBs) have transformed cancer therapy by disrupting inhibitory signals that typically weaken robust anti-tumor immune responses[1]. Despite the success of ICBs, a significant subset of patients remain unresponsive to ICBs owing to various immuno-resistances, which are often propagated by cancer cells[2]. The exploration of combinational therapies involving novel immunomodulatory agents with anti-PD-1/PD-L1 has emerged as a promising approach to overcome intrinsic or acquired resistance to ICBs[3].

Receptor-interacting protein kinase 1 (RIPK1) regulates cell fate through its kinase-dependent and -independent functions and controls proinflammatory responses downstream of multiple innate immune pathways, including those initiated by tumor necrosis factor-α (TNF-α), toll-like receptor ligands, and interferons (IFNs)[4]. Recent studies have shown that genetic knockout (KO) of RIPK1 in cancer cells significantly sensitizes tumors to anti-PD1, leading to drastic changes in the tumor microenvironment (TME), including increased infiltration of effector T cells, reduction of immunosuppressive myeloid cells, and enhanced immunostimulatory cytokine secretion[5–7]. Notably, RIPK1-mediated ICB

resistance requires ubiquitin scaffolding function through its intermediate domain instead of its kinase function. Genetic depletion of RIPK1, but not inactivation of its kinase domain, sensitizes B16F10 tumors to ICBs[5,6]. Hence, targeting RIPK1 scaffolding functions holds promise as a strategy to synergize with ICBs to promote antitumor immunity.

While all RIPK1 inhibitors developed thus far have focused on inhibiting kinase function for the treatment of autoimmune, inflammatory, and neurodegenerative diseases[8], the development of inhibitors specifically targeting the intermediate domain of RIPK1 remains challenging due to the absence of a well-defined binding pocket within this domain. A proteolysis-targeting chimera (PROTAC) is a heterobifunctional molecule that binds both a targeted protein and an E3 ubiquitin ligase to facilitate the formation of a ternary complex, leading to ubiquitination and ultimate degradation of the target protein[9–11].

In this work, we use PROTAC technology to develop LD4172, a highly potent and specific RIPK1 degrader, aimed at abolishing the

[1]The Verna and Marrs McLean Department of Biochemistry and Molecular Pharmacology, Baylor College of Medicine, Houston, TX, USA. [2]Center for NextGen Therapeutics, Baylor College of Medicine, Houston, TX, USA. [3]Department of Molecular and Cellular Biology, Baylor College of Medicine, Houston, TX, USA. [4]Department of Experimental Therapeutics, The University of Texas M.D. Anderson Cancer Center, Houston, TX, USA. [5]Howard Hughes Medical Institute, Janelia Research Campus, Ashburn, VA, USA. [6]Department of Biology and Biochemistry, University of Houston, Houston, TX, USA. [7]James P. Allison Institute, The University of Texas M.D. Anderson Cancer Center, Houston, TX, USA. [8]These authors contributed equally: Xin Yu, Dong Lu. ✉e-mail: dong.lu@bcm.edu; wangj@bcm.edu

                                      

scaffolding functions of RIPK1. LD4172 induces potent and specific RIPK1 degradation and sensitizes multiple preclinical cancer models to anti-PD1 therapy.

## Results

### Development of RIPK1 degrader LD4172
To develop RIPK1 PROTACs, we tested two types of RIPK1 binders: type II RIPK1 inhibitor 1 (also referred as T2I), which targets both the ATP-binding pocket and the allosteric hydrophobic back pocket[12], and type III RIPK1 inhibitor 2, which only binds the hydrophobic back pocket of the kinase domain[13]. To identify the ideal attachment sites for PROTAC linkers, we performed molecular docking of 1 with RIPK1, which revealed a solvent-exposed ethyl group in the 7H-pyrrolo[2,3-d] pyrimidine ring (Fig. 1A). The co-crystal structure of RIPK1 in complex with 2 (PDB: 6R5F) showed that the oxadiazole moiety in 2 was solvent-exposed, providing an ideal exit vector for linker attachment (Fig. 1B).

To identify an appropriate E3 ligase pair for RIPK1 degradation, we synthesized a small library by conjugating RIPK1 binders 1 and 2 to ligands for different E3 ligases, including Cereblon, von Hippel-Lindau tumor suppressor (VHL), murine double minute 2 (MDM2), and a hydrophobic adamantane tag (Fig. 1C). As shown in Fig. 1D–E, PRO-TACs formed by conjugating type II inhibitor 1 to a VHL ligand-induced the most efficient degradation of RIPK1 in Jurkat cells.

We further optimized RIPK1 PROTACs through linker lengths ranging from 2 to 14 methylene groups (Fig. 1F). We found that PROTACs with linker lengths of more than six methylenes were able to effectively degrade >90% of RIPK1 at $1\,\mu M$ after 24 h incubation in Jurkat cells, showing a monotonic trend (Fig. 1F, G). Consistently, the PROTACs exhibited maximal degradation with an 8- to 10-methylene linker and significantly reduced potency with either shorter or longer linkers in B16F10 mouse melanoma cells (Fig. 1F, G). Considering the potency of both human and mouse cells, we chose a combination of a type II RIPK1 binder, a VHL ligand, and a 10-methylene linker as the lead RIPK1 degrader, designated as LD4172 (Fig. 2A).

### LD4172 induces potent RIPK1 degradation in vitro
LD4172 induced potent RIPK1 degradation (concentration to induce 50% protein degradation $DC_{50} = 4$–$400\,nM$) in a panel of human and mouse cancer cell lines (Fig. 2B, C, Supplementary Fig. 1). To investigate the kinetics of LD4172-induced RIPK1 degradation and resynthesis rates, Jurkat and B16F10 cells were treated with LD4172 for different time points, followed by washout after 24 h. With $1\,\mu M$ LD4172 treatment, >90% of RIPK1 was degraded within 2 h and 4 h in Jurkat and B16F10 cells, respectively (Fig. 2D, E). Four hours after the removal of LD4172, RIPK1 starts to resynthesize in both cell lines. The resynthesis half-lives are ~48 h and 24 h in Jurkat and B16F10 cells, respectively (Fig. 2D, E). Collectively, these data demonstrated that LD4172 is a potent RIPK1 degrader with rapid and sustained effects in vitro.

### LD4172 engages RIPK1 and forms a ternary complex
To elucidate the formation of a binary complex during RIPK1 degradation, we developed a competitive Nano-Bioluminescence Resonance Energy transfer (NanoBRET)-based target engagement (TE) assay to quantify the binding between RIPK1 and LD4172 in cells[14–16]. First, we developed a RIPK1 tracer by conjugating the LD4172 warhead T2I with a BODIPY-590 fluorescent dye, dubbed T2-590[17]. The dissociation equilibrium constant ($K_d$) between the tracer and RIPK1 was determined to be $0.5\,\mu M$ by titrating the tracer in HEK293T cells expressing a nLuc-RIPK1 fusion protein. Subsequently, with the tracer concentration at its $K_d$ value, LD4172 competed with the tracer with an $IC_{50}$ value of $3.7\,\mu M$ (Fig. 2F). Based on the Cheng-Prusoff equation, the apparent $K_i$ between LD4172 and RIPK1 in the cells was $1.9\,\mu M$. Using a recombinant human RIPK1 protein, we measured the biochemical $K_i$ between LD4172 and human RIPK1 to be $4.8\,nM$ (Fig. 2G), which is 395 folds smaller than the corresponding $K_i$ in cells. This is usually expected,

considering that the large molecular weight of LD4172 may lead to poor cellular permeability. However, the fact that the $DC_{50}$ values of LD4172 are much smaller than its TE $IC_{50}$ values demonstrates the sub-stoichiometric degradation of RIPK1 induced by LD4172. Additionally, we synthesized an LD4172 negative control (LD4172-NC, also referred to as NC, Fig. 2A) using a VHL ligand diastereomer that does not bind to VHL. As expected, LD4172-NC showed TE similar to that of LD4172 (Fig. 2F).

To test whether LD4172 induces ternary complex formation with RIPK1 and VHL, we co-transfected HEK293 cells with nLuc-RIPK1 and VHL-Halo labeled with BODIPY-590 dye. The addition of LD4172, but not LD4172-NC, induced NanoBRET between RIPK1 and VHL, demonstrating the formation of a ternary complex among {RIPK1-LD4172-VHL} (Fig. 2H).

### LD4172 degrades RIPK1 with high specificity through UPS
The mechanistic action of PROTACs involves bringing the protein of interest (POI) into close proximity to an E3 ligase, which ubiquitinates the POI for degradation by the proteasome. To confirm that LD4172 functions through ubiquitin proteosome system (UPS), we disrupted ternary complex formation by introducing an excess of RIPK1 or VHL ligands, which led to attenuation of RIPK1 degradation induced by LD4172. Moreover, blocking Cul2 E3 ligase with the neddylation inhibitor MLN4924 or inhibiting the proteasome with carfilzomib reversed the potent degradation of RIPK1 by LD4172 in both Jurkat and B16F10 cells (Fig. 2I). These findings indicate that LD4172 induces protein degradation through ternary complex formation and the UPS machinery.

The RIPK1 binder used in LD4172 is a typical type II kinase inhibitor bound to some off-target kinases, including TrkA, Flt1, Flt4, Ret, Met, Mer, Fak, FGFR1, and MLK1[12]. To evaluate the specificity of LD4172, we performed mass spectrometry (MS) analysis of the whole cellular proteome. Because Jurkat and B16F10 cells lack expression of all the aforementioned off-target kinases, MDA-MB-231 cells were chosen and treated with either LD4172 (200 nM) or LD4172-NC (200 nM) for 6 h. Among the >10,000 proteins detected, RIPK1 was the only protein degraded by LD4172 (the red dot in Fig. 2J), and no degradation of off-target kinases was observed (blue dots in Fig. 2J). Additionally, at effective LD4172 concentrations (16 nM–10 μM) that induce RIPK1 degradation in THP1 cells, we observed no significant changes in the protein levels of related kinases, such as RIPK2 and RIPK3 (Supplementary Fig. 2). These findings are consistent with previous studies showing that PROTACs with promiscuous target protein binders can achieve enhanced selectivity through protein-protein interactions with the E3 ligase involved[18].

### LD4172 boosts apoptosis in TNFα-stimulated B16F10 cells
RIPK1-deficient MEFs exhibit a severe impairment in their ability to activate NF-κB signaling, whereas kinase-dead RIPK1 knock-in mice remain viable and display normal TNFR1-mediated NF-κB signaling[19–22]. Moreover, unlike the kinase-dead RIPK1 scenario, genetic deletion of RIPK1 has been shown to trigger apoptosis and necroptosis both in vitro and in vivo[23–28]. To evaluate NF-κB activity in B16F10 cells, we transiently transfected them with a plasmid encoding a Nanoluc reporter containing the NF-κB response element, allowing us to monitor NF-κB activity through luminescence. Upon TNF-α treatment, B16F10 cells exhibited a robust induction of NF-κB activity, which was significantly attenuated by LD4172 but not by the RIPK1 kinase inhibitor (Fig. 3A). To investigate cellular death in the B16F10 mouse melanoma cell model and its correlation with the mechanism of RIPK1 downregulation rather than kinase inhibition, we employed various tool molecules, including LD4172, T2I, TNFα, Smac mimetic LCL161, and the pan-caspase inhibitor Z-VAD-FMK. Although B16F10 cells remained unresponsive to necroptotic triggers (TNFα + LCL161 + Z-VAD-FMK, Supplementary Fig. 3), likely due

                                                      

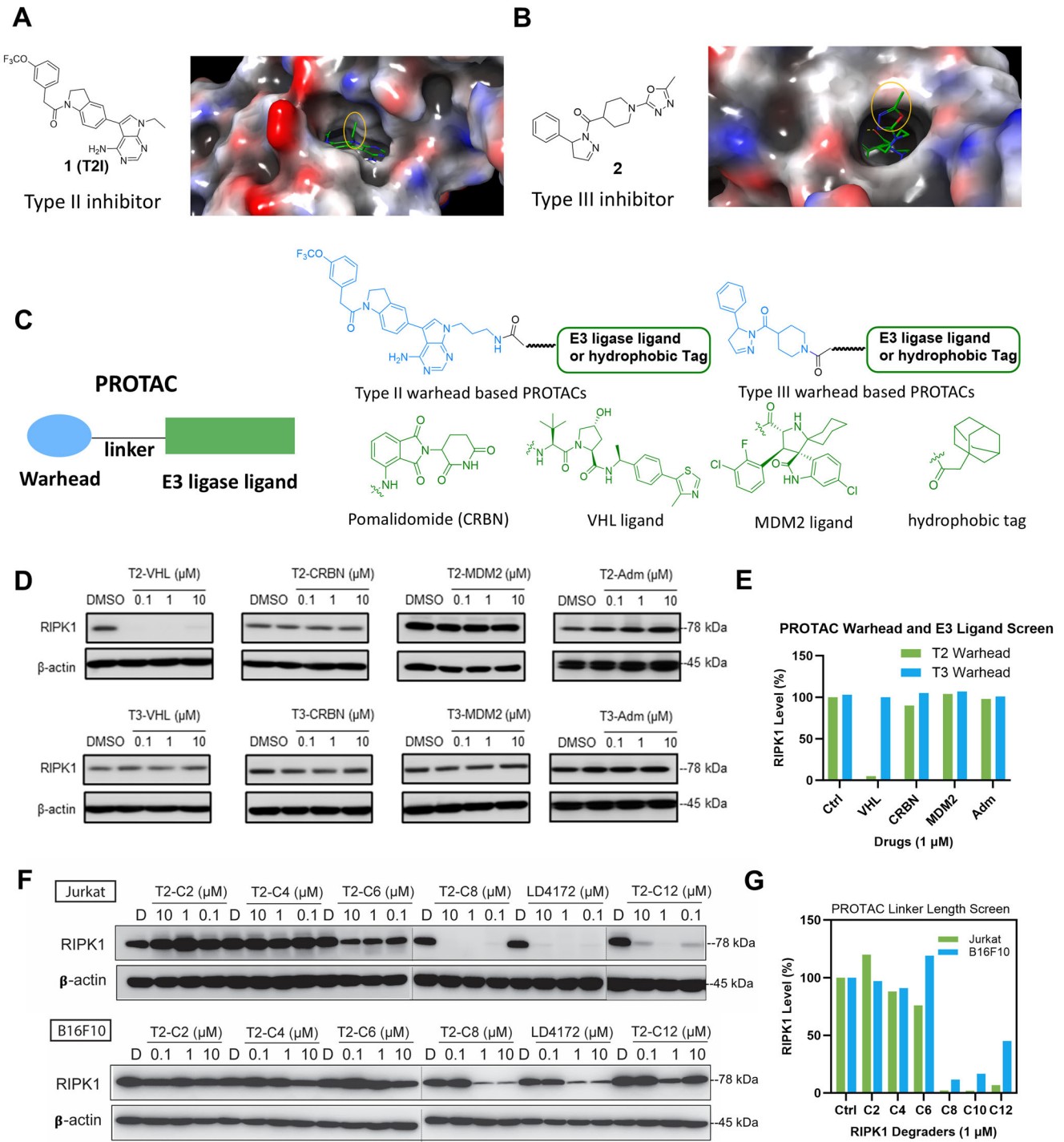

**Fig. 1 | Design and screen of RIPK1 PROTACs. A** Chemical structure and docking model of Type II inhibitor 1 bound to the kinase domain of RIPK1 (PDB: 4NEU). The solvent-exposed group is highlighted in a yellow circle. **B** Chemical structure and co-crystal structure of Type III inhibitor 2 in complex with RIPK1 (PDB: 6R5F). The solvent-exposed group of inhibitor 2 is highlighted in a yellow circle. **C** Small library design of RIPK1 PROTACs. Type II inhibitor 1 or Type III inhibitor 2 was conjugated with various E3 ligase ligands or an adamantane tag to generate a small library of RIPK1 PROTACs. **D** Quantification of RIPK1 levels in Jurkat cells treated with the indicated compounds for 24 h at 0, 0.1, 1, and 10 μM, followed by Western blot analysis. This preliminary screening experiment was performed once for all compounds, except for T2-VHL, which was tested in three independent experiments. Source data are provided as a Source Data file. **E** Quantification of RIPK1 levels from (**D**). **F** Quantification of RIPK1 levels in Jurkat and B16F10 cells treated with Type II inhibitor-based PROTACs with varying linker lengths for 24 h, followed by Western blot analysis. The optimal linker length was identified in this single experiment. Source data are provided as a Source Data file. **G** Quantification of RIPK1 levels from (**F**).

to the low expression of RIPK3[29], the combination of TNFα and LD4172 induced significant apoptosis (Fig. 3B–D). This was evidenced by the enhanced surface exposure of phosphatidylserine (Fig. 3B) and increased levels of cleaved caspase-3/7 and cleaved PARP (Fig. 3C, D). Notably, these apoptotic effects were reversed with

Z-VAD-FMK treatment (Fig. 3B–D). In contrast, inhibition of RIPK1 kinase activity by T2I did not trigger TNFα-mediated apoptosis (Fig. 3B–D). Additionally, apoptosis induced by LD4172 plus TNFα involves membrane ruptures, as indicated by the enhanced production of ATP in extracellular environments (Fig. 3E), loss of nuclear

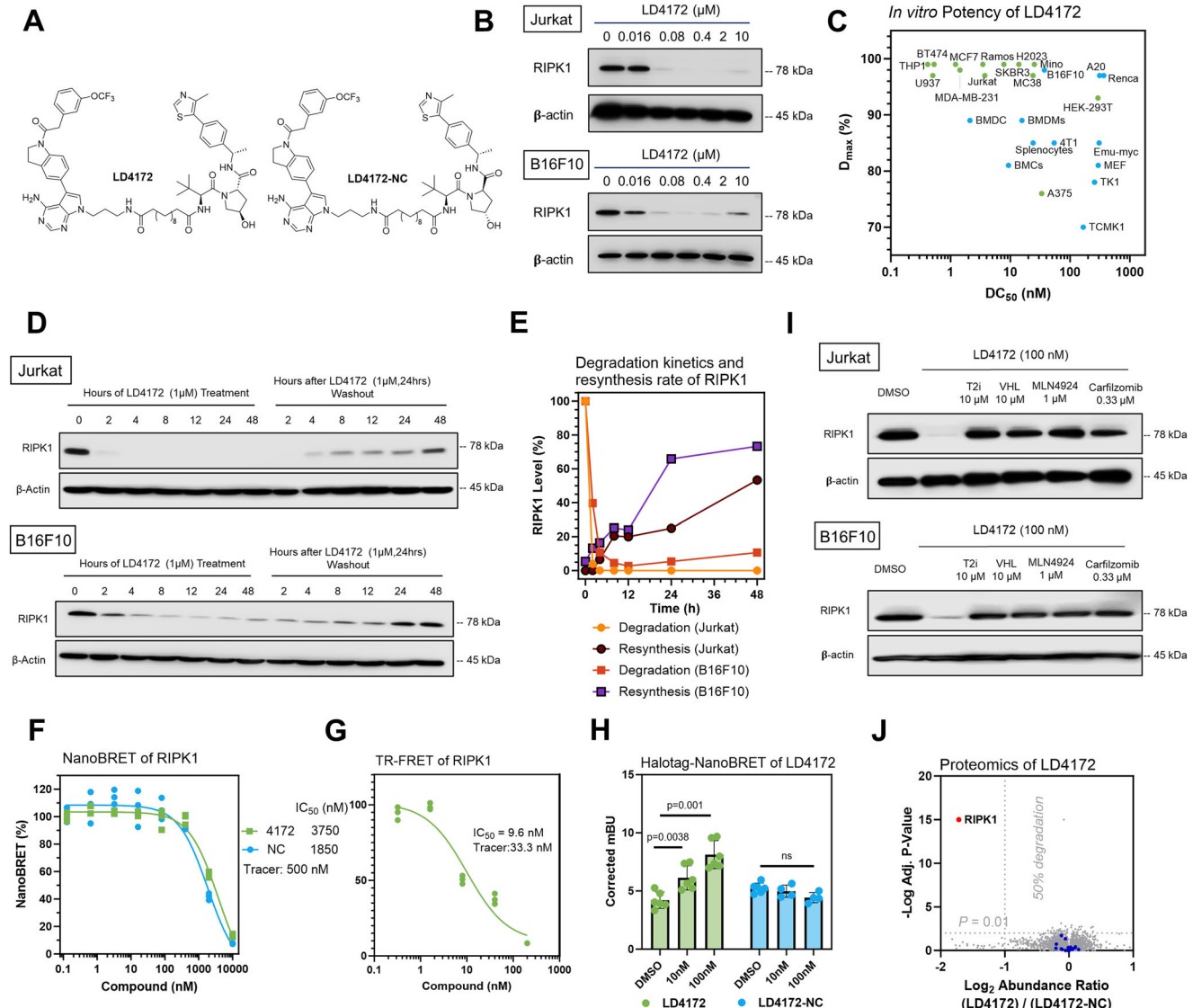

**Fig. 2 | LD4172, a RIPK1 PROTAC, induces potent and highly specific degradation of RIPK1 in a panel of cell lines. A** Chemical structures of RIPK1 PROTAC LD4172 and its negative control, LD4172-NC. LD4172-NC shares the same warhead and linker as LD4172 but contains an inactive VHL ligand, making it unable to recruit VHL for ubiquitination. **B** Quantification of RIPK1 levels in Jurkat and B16F10 cells treated with various concentrations of LD4172 for 24 h, analyzed by Western blot. Results are representative of three independent experiments. Source data are provided as a Source Data file. **C** The $DC_{50}$ and $D_{max}$ values of LD4172 were determined in various human and mouse cell lines. The screening of LD4172 across different cell types was conducted in a single experiment. $DC_{50}$ represents the concentration at which 50% of RIPK1 is degraded, while Dmax indicates the maximum level of degradation achieved. **D** Kinetics of RIPK1 degradation induced by LD4172 (1 μM) and resynthesis upon LD4172 washout in Jurkat and B16F10 cells. Representative Western blots ($n = 3$) show that the degradation half-life of RIPK1 is less than 2 h in both cell lines. RIPK1 resynthesis begins 4 h post-washout, with half-lives of ~48 h in Jurkat cells and ~24 h in B16F10 cells. Source data are provided as a Source Data file. **E** Quantification of RIPK1 degradation and resynthesis from (**D**). **F** NanoBRET-based in-cell RIPK1 target engagement assay. HEK293 cells transfected with nLuc-RIPK1 were incubated with a RIPK1 NanoBRET tracer (500 nM) and different concentrations of LD4172 or LD4172-NC ($n = 3$ biological independent replicates, three independent experiments). **G** Time-resolved fluorescence resonance energy transfer (TR-FRET) biochemical binding assay for RIPK1. GST-tagged

human RIPK1 (1 nM), Tb-labeled anti-GST antibody (0.3 nM), a RIPK1 TR-FRET tracer (350 nM), and various concentrations of LD4172 were incubated for 2 h ($n = 3$ biological independent replicates, three independent experiments), followed by TR-FRET measurements at excitation 340 nm and emission 495/520 nm. **H** NanoBRET-based ternary complex formation assay. HEK293T cells co-transfected with nLuc-RIPK1 and VHL-HaloTag were treated with different concentrations of LD4172 or LD4172-NC ($n = 3$ biological replicates, three independent experiments). Graph bars represent mean ± SD, and statistical significance was determined using a two-tailed unpaired $t$-test with P values indicated. **I** LD4172-induced RIPK1 degradation depends on ternary complex formation, neddylation, and proteasome activity. Representative Western blots (three independent experiments) of RIPK1 in Jurkat and B16F10 cells treated with T2I, a VHL ligand, MLN4924 (neddylation inhibitor), or Carfilzomib (proteasome inhibitor) for 4 h, followed by LD4172 treatment. Source data are provided as a Source Data file. **J** Proteomic profiling of LD4172-induced degradation. MDA-MB-231 cells were treated with LD4172 or LD4172-NC (200 nM) for 6 h ($n = 3$ biological independent replicates from the single experiment). Proteins were ranked in a volcano plot based on their P value (−log10) and fold change ($log_2$ FC) between LD4172 and LD4172-NC treatments. RIPK1 (red dot) showed >50% degradation with $P < 0.01$, while blue dots represent kinases inhibited by the LD4172 warhead but not degraded. Source data are provided as a Source Data file.

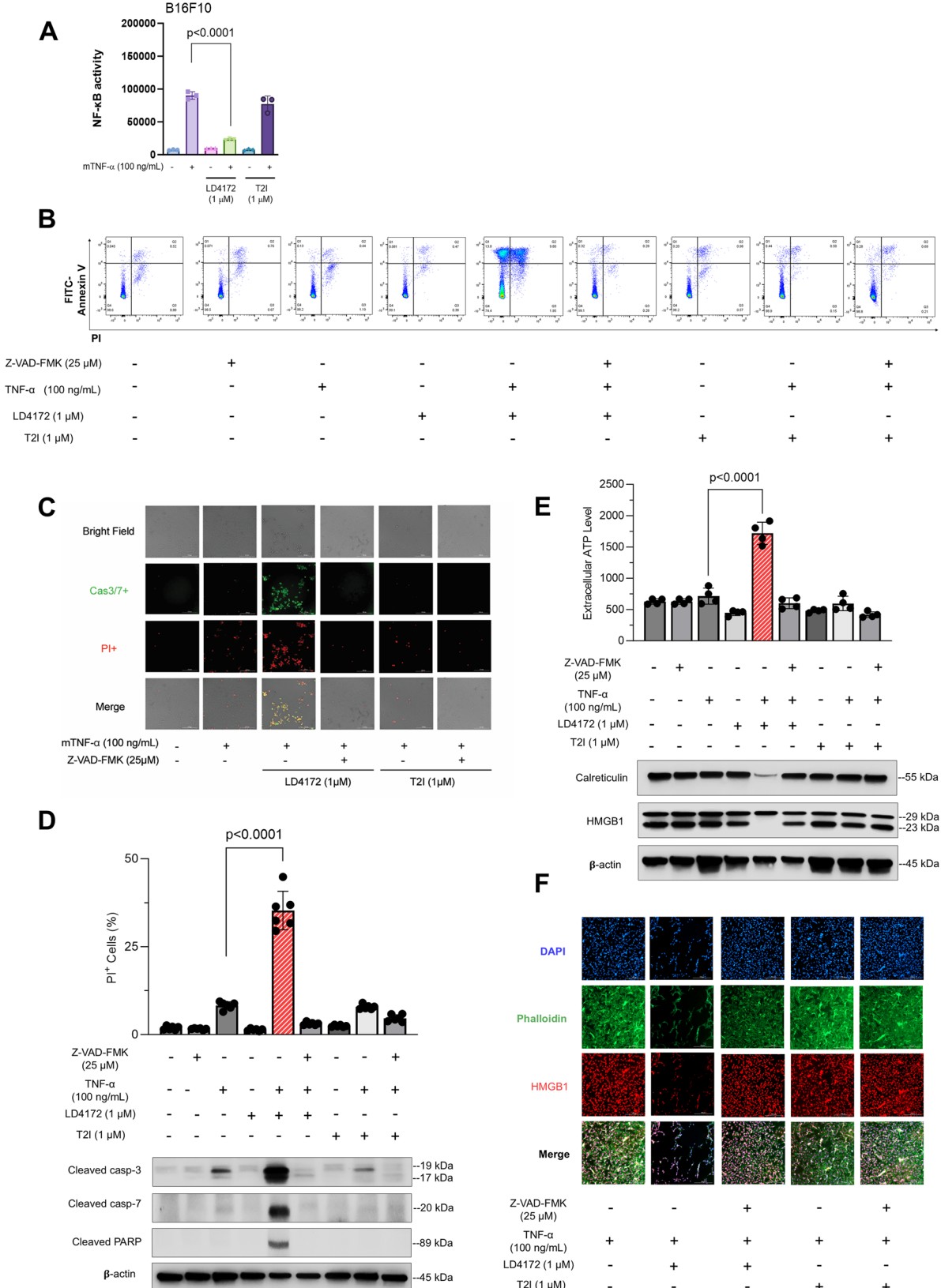

HMGB1 (High Mobility Group Box 1, Fig. 3E, F) and downregulated calreticulin (Fig. 3E).

## Pharmacokinetic and pharmacodynamic study of LD4172

LD4172 has half-lives of 21.1 and 9.7 min in human and mouse liver S9 fractions, respectively, corresponding to intrinsic clearance ($CL_{int}$) of 32.8 and 71.6 $\mu L \cdot min^{-1} \cdot mg^{-1}$ protein. In human primary hepatocytes, the half-life of LD4172 is 56.3 min, which corresponds to a predicted $CL_{int}$ of 15.6 $mL \cdot min^{-1} \cdot kg^{-1}$ in humans. It should be noted that the predicted intrinsic clearance in primary hepatocytes was >2000 times slower than that in liver S9 fractions, possibly due to the low membrane permeability of

**Fig. 3 | LD4172 sensitizes B16F10 cells to TNFα-mediated apoptosis.** B16F10 cells were treated with or without TNFα (100 ng/mL), Z-VAD-FMK (25 μM), LD4172 (1 μM), and/or T2I (1 μM) for 72 h. **A** NF-κB activity in B16F10 cells expressing a NanoLuc reporter for NF-κB response. Data represent the mean NF-κB activity ± SD ($n = 3$ biologically independent samples from two independent experiments). Statistical significance was determined using a two-tailed unpaired $t$-test, with $P$ values indicated. **B** Representative flow cytometry dot plots showing apoptosis in B16F10 cells from three independent experiments ($n = 3$ biologically independent samples per experiment). Viable cells (FITC−/PI−) are located in the lower left quadrant, early apoptotic cells (FITC+/PI−) in the lower right quadrant, and late apoptotic cells (FITC+/PI+) in the upper right quadrant. **C** Representative images of B16F10 cells stained with PI (red) and caspase 3/7 (green). Data are shown from three independent experiments ($n = 3$ biologically independent samples per experiment). Cell death was quantified by PI uptake using the Cytation 5 imager. **D** Top:

Percentage of PI+ B16F10 cells after 72 h of the indicated treatments, calculated as PI+ (%) = (Number of PI+ cells/Total cells) × 100%, using the Cytation 5 imager. Data represent the mean ± SD ($n = 6$ biologically independent replicates from three independent experiments). Statistical significance was determined using a two-tailed unpaired $t$-test, with $P$ values indicated. Bottom: Western blots showing expression of cleaved caspase-3, cleaved caspase-7, and cleaved PARP in B16F10 cells from one experiment. Source data are provided as a Source Data file. **E** Top: Quantification of extracellular ATP levels secreted by B16F10 cells. Data represent mean extracellular ATP levels ± SD ($n = 4$ biologically independent samples from one experiment), with statistical significance determined by a two-tailed unpaired $t$-test, and $P$ values indicated. Bottom: Representative Western blots showing expression of HMGB1 and calreticulin in B16F10 cells. Source data are provided as a Source Data file. **F** Immunofluorescence staining of HMGB1 in B16F10 cells, with data from three biologically independent samples from one experiment.

LD4172, which protects it from being metabolized (Supplementary Table 1).

Next, we evaluated the pharmacokinetics (PK) of LD4172 in C57BL/6J (B6) mice (Fig. 4A). With 1 mg/kg intravenous (i.v.) administration in C57BL/6J mice, LD4172 showed a half-life ($t_{1/2}$) of $3.3 ± 2.1$ h, a maximum plasma concentration ($C_{max}$) of $6.3 ± 0.8$ μM, and an area under the concentration-time curve (AUC) of $0.7 ± 0.07$ μM·h. The volume of distribution ($V_d$) of LD4172 was $1100 ± 200$ mL·kg$^{-1}$, which is much greater than the mouse plasma volume (77–80 mL·kg$^{-1}$), suggesting that LD4172 has strong affinities to tissues. The clearance of LD4172 is $19.8 ± 1.8$ mL·min$^{-1}$·kg$^{-1}$.

Intraperitoneal (i.p.) administration of LD4172 (10 mg/kg) to C57BL/6J mice led to $C_{max}$, $t_{1/2}$, and AUC as 2.9 μM, 1.5 h, and 2.7 μM·h, respectively (Fig. 4A, Supplementary Table 1). Considering an AUC of 0.7 μM·h for i.v. administration (1 mg/kg), i.p. administration of LD4172 achieved 39% bioavailability.

To investigate the pharmacodynamics of LD4172 in vivo, we administered LD4172 via the i.p. route and observed a 60% reduction in RIPK1 levels in tumors (20 mg/kg, b.i.d., i.p.) (Fig. 4B, C). In contrast, less than 50% RIPK1 degradation was observed in the spleen, and no significant RIPK1 degradation was observed in other organs, including the lymph nodes, PBMCs, lungs, and bone marrow (Fig. 4B, C).

The hERG channel inhibition assay is a commonly used safety assay to identify compounds that exhibit cardiotoxicity related to hERG inhibition in vivo. LD4172 exhibited no obvious inhibition of hERG, even at 30 μM (Supplementary Table 1), indicating that LD4172 has a good safety margin for hERG inhibition.

### LD4172 sensitizes tumors to anti-PD1 therapy

Utilizing CRISPR-Cas9 technology, we generated RIPK1-KO B16F10 cells and implanted them into mice to examine their response to anti-PD1 treatment (Fig. 4D, E). Align with previous reports[5–7], our findings demonstrated that tumors lacking RIPK1 exhibit heightened sensitivity to anti-PD1 treatment (Fig. 4E). Subsequently, we explored whether pharmacological degradation of RIPK1 could replicate the effects observed in RIPK1-null B16F10 tumors. Consistent with the genetic study, mice treated with anti-PD1 or LD4172 alone showed tumor progression similar to that of the untreated mice. However, LD4172 sensitized B16F10 tumors to anti-PD1 therapy (Fig. 4F–I), with long-term administration of LD4172 showing no impact on mouse body weight (Fig. 4J). To test whether inhibition of RIPK1 kinase activity also enhances tumor responses to ICB therapy, we treated B16F10 xenograft tumors with the RIPK1 kinase inhibitor T2I, alone or in combination with anti-PD1. Unlike the RIPK1 degrader LD4172, the RIPK1 kinase inhibitor T2I failed to sensitize B16F10 tumors to anti-PD1 treatment (Fig. 4K).

We also tested a syngeneic MC38 colon cancer model, which exhibited a limited response to anti-PD1 treatment. Consistent with the B16F10 tumor model, LD4172 sensitized MC38 tumors to anti-PD1 therapy (Supplementary Fig. 4A, B).

### LD4172 triggers immunogenic cell death in B16F10 tumor

To understand the observed synergistic effects of LD4172 and anti-PD1, we administered vehicle, LD4172, anti-PD1, or a combination of LD4172 and anti-PD1 in C57BL/6J mice with B16F10 tumors for a short duration. A 5-day treatment with LD4172 was sufficient to induce substantial degradation of RIPK1 in the tumor (Fig. 5A, 1st column). LD4142 treatment significantly disrupted the dense structure of B16F10 tumors, as evidenced by a marked reduction in cellular density observed in H&E staining (Fig. 5A, 2nd column). Importantly, a notable increase in cleaved caspase 3/7 levels was observed in the LD4172-treated tumors, indicating the occurrence of apoptosis (Fig. 5A, 3rd and 4th columns). Supporting the activation of immunogenic apoptosis, we observed a significant increase in plasma HMGB1 levels (Fig. 5B) and enhanced exposure of calreticulin on the surface of B16F10 tumor cells (Fig. 5A, 5th column).

### LD4172 enhances anti-tumor immunity

To elucidate how the combination of LD4172 plus anti-PD1 promotes anti-tumor immunity, multiparameter flow cytometry was employed to evaluate tumor-infiltrating lymphocytes (TILs) within the TME of mice receiving different treatments (Supplementary Fig. 5). Initially, we confirmed the successful blockade of PD1 on T cells (CD8 + PD1+) with an anti-PD1 antibody (Fig. 5C). LD4172-induced immunogenic cell death (ICD) led to a notable expansion of CD4+ T cells (Fig. 5A, 7th column, and 5D), conventional dendritic cells (cDC1, CD45+CD11C +IAIE+XCR1+, Fig. 5E), and macrophages (CD45+CD11b+F4/80+, Fig. 5A, 8th column, and 5F) within the TME, all of which contribute to antigen presentation and cytotoxic T cell priming and activation. Additionally, combined therapy with LD4172 and anti-PD1 significantly enhanced anti-PD1 positivity in immunologically cold B16F10 tumors, as demonstrated by increased infiltration of cytotoxic CD8+ T cells (CD8+IFN-γ+, Fig. 5A, 6th column, and 5G-H) and decreased infiltration of FOXP3+ T regulatory cells (Fig. 5A, 7th column) within the TME. To confirm the contribution of CD8+ T cells to the antitumor effect, we conducted a CD8+ T cell depletion experiment, revealing that the synergy between anti-PD1 and LD4172 was nullified in the absence of CD8+ T cells (Fig. 5I). Results from the cytokine array profiling of plasma further supported synergistic effects of combined treatment, showing a significant enhancement in the production of immune cell proliferation cytokines, including IFN-γ and IL2 (Fig. 5J).

### Discussion

RIPK1 is a critical regulator involved in cellular processes and proinflammatory signaling and exerts its effects through both kinase-dependent and kinase-independent mechanisms. In particular, its ubiquitin scaffolding function through K376 has been implicated in conferring intrinsic and extrinsic resistance to ICB and is a potential target for cancer immunotherapy[5,6]. However, the development of inhibitors that specifically target the intermediate scaffolding domain is challenging because of the absence of a well-defined

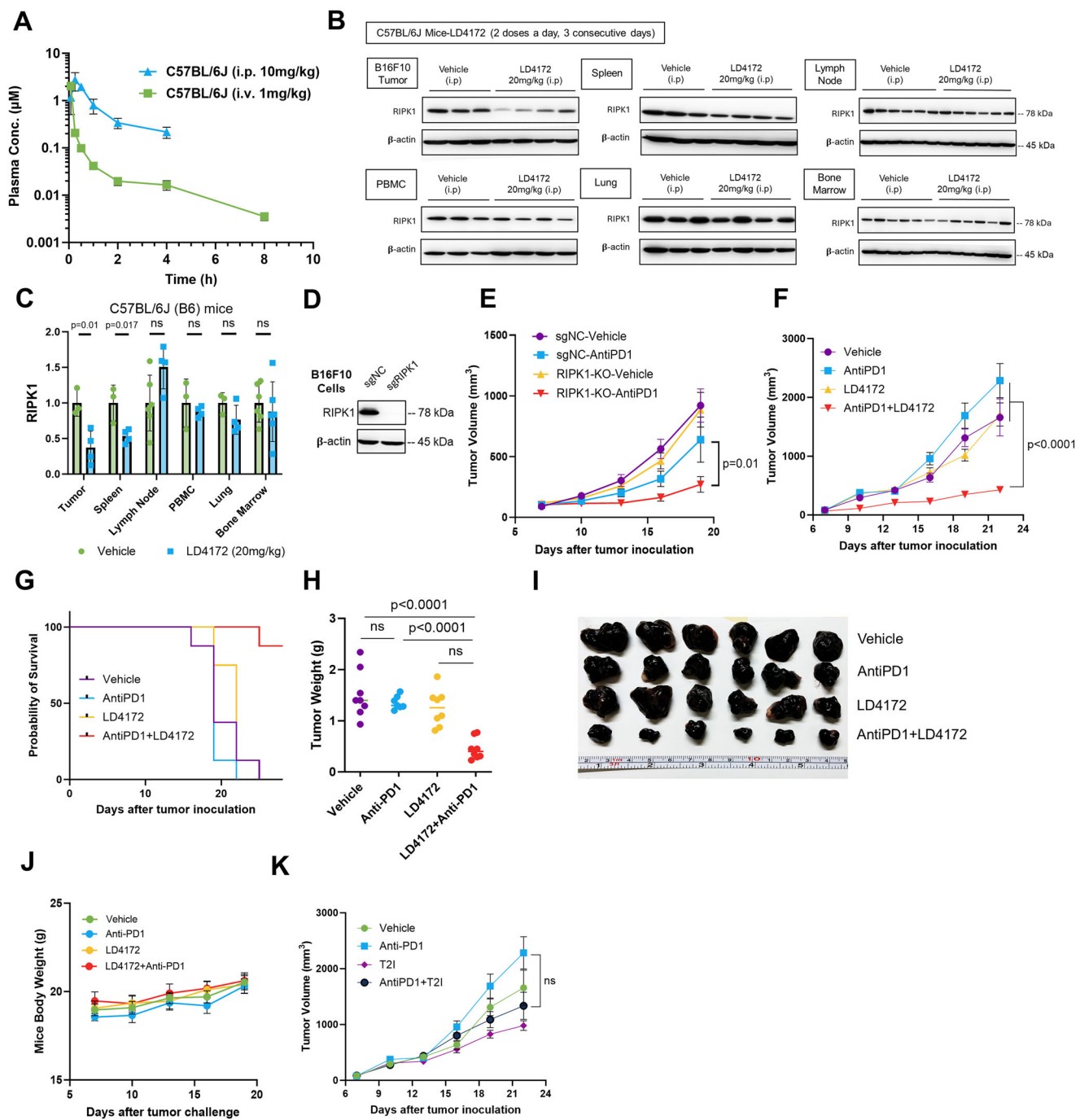

binding pocket. In this study, we used PROTAC technology to address this limitation and successfully developed a RIPK1 degrader, LD4172, with potent and specific RIPK1 degradation both in vitro and in vivo.

Upon TNF-α signaling, RIPK1 is recruited to complex I, where it acts as a crucial scaffold, essential for the ubiquitylation-dependent activation of NF-κB signaling. Our findings demonstrate that LD4172-induced RIPK1 degradation, unlike the action of RIPK1 kinase inhibitors, significantly impairs TNF-α-induced NF-κB activation. Furthermore, the combination of TNF-α and LD4172 markedly induces apoptosis in B16F10 cells. This suggests that acute RIPK1 deletion shifts the cellular response from a pro-survival to a pro-death state by altering the composition of complexes I and II. Interestingly, the Meier group recently developed a similar RIPK1 degrader[21], revealing that RIPK1 degradation promotes the formation of a RIPK1-independent complex I. This complex displays enhanced interactions among

TNFR1, TRADD, TRAF2, cIAP, and HOIP, leading to deregulated TNF signaling through increased ubiquitylation. These findings underscore the potential of RIPK1 degraders as valuable chemical tools for in vitro studies, offering insights into the biological roles of RIPK1 within various signaling complexes.

LD4172 has also demonstrated significant therapeutic efficacy in vivo by inducing RIPK1 degradation within tumors and exhibiting a synergistic effect on tumor growth inhibition when combined with anti-PD1 therapy. Although LD4172 plays a role in inducing ICD in B16F10 tumors, the degradation of RIPK1 alone is insufficient, as RIPK1 primarily acts as a brake on immunogenic pathways[21]. Therefore, additional ligands are required to fully activate these pathways. Anti-PD1 can supply these necessary ligands by promoting TNF production, thereby sensitizing cancer cells to cell death in the absence of RIPK1. This synergistic relationship is further evidenced by the finding that the combination of LD4172 and anti-PD1 loses its efficacy when anti-

**Fig. 4 | LD4172 synergizes with Anti-PD1 to inhibit tumor growth. A** Plasma concentrations of LD4172 in C57BL/6J mice administered 1 mg/kg intravenously (i.v.) or 10 mg/kg intraperitoneally (i.p.) (*n* = 4 mice/group). **B** Representative immunoblots showing RIPK1 levels in various tissues from C57BL/6J mice treated with vehicle or LD4172. Tissues analyzed include B16F10 tumor, spleen, PBMC, lung (*n* = 3 mice for vehicle, *n* = 4 mice for LD4172), as well as lymph node and bone marrow (*n* = 6 mice for both groups). Source data are provided as a Source Data file. **C** Densitometric analysis of RIPK1 protein levels in various tissues based on immunoblot images from (**B**). Tissues analyzed: B16F10 tumor, spleen, PBMC, lung (vehicle *n* = 3 mice, LD4172 *n* = 4 mice), lymph node, and bone marrow (*n* = 6 mice for both groups). Graph bars represent mean values ± SD and statistical significance was calculated with a two-tailed unpaired *t*-test and *P* values are indicated. **D** Representative immunoblots showing RIPK1 levels in B16F10 cells transduced with either RIPK1-specific gRNA or a non-targeting control (sgNC). Data represent results from three independent experiments (*n* = 3 biologically independent samples per experiment). **E** B16F10-RIPK1-KO tumors sensitized to anti-PD1 treatment: $3 \times 10^5$ B16F10 cells transduced with gRNA specific for RIPK1 or non-targetable control (sgNC) were inoculated into C57BL/6J mice. After 7 days, mice with measurable tumors (~100 mm³) were randomly treated with or without anti-PD1 in vivo (100 µg per dose, i.p., every 3 days, *n* = 8 mice/group). Each symbol represents the mean tumor volume, with error bars indicating SEM. Statistical analysis was performed using two-way ANOVA followed by Sidak's multiple comparisons test, with significance levels indicated. Results are representative of three independent experiments. **F** Tumor growth curve of mice with B16F10 tumors treated with LD4172 and/or anti-PD1 (*n* = 8 mice/group). C57B6/J mice were subcutaneously inoculated with $3 \times 10^5$ B16F10 tumor cells. After 7 days (tumor size ~100 mm³), mice were treated every 3 days with anti-PD1 (100 µg per dose, i.p.), daily with LD4172 (20 mg/kg, i.p.), a combination of LD4172 and anti-PD1 (same dose as their individual doses), or their corresponding vehicle control. Each symbol represents the mean tumor volume, with error bars indicating SEM. Statistical analysis was performed using two-way ANOVA followed by Sidak's multiple comparisons test, with significance levels indicated. Results are representative of three independent experiments. **G** Kaplan-Meier survival curve for all experimental groups. **H** Final tumor weight (g) from (**F**) after 22 d of treatment (*n* = 8 mice/group). Statistical significance was calculated with a two-tailed unpaired *t*-test and *P* values are indicated. **I** Representative images of B16F10 tumors collected at the end of treatment. **J** Mouse body weight (*n* = 8 mice/group). **K** Tumor growth curve of mice with B16F10 tumors treated with T2I and/or anti-PD1 (*n* = 8 mice/group). The experimental conditions and treatment regimens were the same as (**F**) except using the RIPK1 kinase inhibitor T2I (20 mg/kg, i.p.) to replace LD4172. Each symbol represents the mean tumor volume, with error bars indicating SEM. Statistical analysis was performed using two-way ANOVA followed by Sidak's multiple comparisons test, with significance levels indicated.

---

TNFα is introduced, completely abolishing their combined effect (Supplementary Fig. 6). Moreover, when used in conjunction with anti-PD1, LD4172 reshapes the tumor immune microenvironment by enhancing the infiltration of dendritic cells and IFNγ+ T cells, as well as by promoting the secretion of immunostimulatory cytokines, leading to antitumor effects.

The suboptimal pharmacokinetic properties of LD4172, particularly its high in vivo clearance and low unbound plasma drug concentration, present challenges for achieving optimal RIPK1 degradation in vivo following intraperitoneal administration. Interestingly, intratumoral administration of LD4172 improved RIPK1 degradation in the tumors (Supplementary Fig. 7), suggesting that the incomplete degradation of RIPK1 in tumors via i.p. injections may be attributed to poor penetration and/or accumulation of LD4172 in tumor tissues. Additionally, LD4172 had poor permeability in cells based on the divergence observed between biochemical and cellular assays for TE (Fig. 2G, H). To enhance the pharmacodynamic properties of LD4172, optimization of medicinal chemistry is necessary to further improve its physicochemical and pharmacokinetic characteristics. Previous studies have reported successful optimization strategies, including optimizing linker structures[30], E3 ligands, and kinase warheads[31], introducing intramolecular hydrogen bonds[32], and converting the drug into a prodrug form[33]. Our future work will implement these optimization approaches to enhance membrane permeability and the overall pharmacokinetic profile of LD4172, thereby improving its potency for RIPK1 degradation in vivo.

As a therapeutic modality, the on-target toxicity profiles of RIPK1 degraders remain unclear. Although mice are a convenient model system for exploring the functions of cellular signaling pathways, human genetics provides the best models of human diseases and guides the selection of new targets for drug discovery[34]. Unlike Ripk1 KO mice, which die within 1–3 days of age due to the critical role of RIPK1 in multiple tissues and organs[35], the phenotypes of homozygous loss-of-function RIPK1 mutations in humans are relatively less severe[36]. Although the permanent loss of RIPK1 in patients leads to severe immunodeficiency and/or intestinal inflammation[37], chemical-induced protein degradation of RIPK1, which is acute, transient, and potentially tissue-specific[38], might be more tolerable in humans. However, the safety profiles of RIPK1 degraders need to be evaluated in future clinical studies. Furthermore, we found that decreasing RIPK1 dosing frequency by 50% resulted in a similar tumor inhibition effect when combined with anti-PD1 (Supplementary Fig. 8), suggesting that the

safety profile of RIPK1 degraders can be further improved by optimizing the dosing regimen.

Importantly, we primarily observed RIPK1 degradation in tumors to a lesser extent in the spleen and no degradation in other organs (Fig. 4B, C). In contrast to small-molecule inhibitors, protein degraders can achieve tissue-specific target degradation by leveraging tissue-specific E3 ligases. One notable example is a BCL-X$_L$ degrader developed by the groups of Zhou and Zheng, which spares platelets due to the low expression of VHL in platelets[39]. However, according to proteinatlas.org, VHL expression is ubiquitous in major organs, which does not explain the tumor-selective RIPK1 degradation induced by LD4172. Albumin, which constitutes approximately 60% of total plasma protein, preferentially accumulates in tumors due to the high demand for amino acids and energy in these tissues[40,41]. Given that 98.6% of LD4172 is bound to plasma proteins (Supplementary Table 1), it is plausible that LD4172 may be "piggybacking" on albumin accumulation in tumors, thereby achieving tumor-selective RIPK1 degradation. Supporting this hypothesis, following a single 20 mg/kg dose of LD4172 in C57BL/6J mice bearing B16F10 tumors, LD4172 was found to persist in the tumor for an extended period compared to other tissues (Supplementary Fig. 9). This prolonged retention in the tumor could further mitigate potential toxicity concerns related to RIPK1 degradation in normal tissues.

In summary, we developed an RIPK1 degrader with high degradation specificity in cells and tumor selectivity in vivo. Our work not only provides a chemical probe to explore the effects of RIPK1 degradation in biology, but also the proof-of-concept study demonstrating that pharmacological degradation of RIPK1 synergizes with anti-PD1 to overcome resistance to ICBs. Considering the predicted safety profile of RIPK1 degradation based on human genetics, we envision that further optimized RIPK1 degraders have the potential to improve cancer immunotherapy.

## Methods

### Cell lines

Human and mouse hematopoietic cell lines, namely Jurkat, Ramos, THP1, U937, TK1, and A20, and mouse melanoma B16F10 cell lines, were procured from ATCC. MC38 colon carcinoma and H2023 lung carcinoma cells were provided by Dr. Weiyi Peng. A375 melanoma cells were acquired from the Cell Core at the MD Anderson Cancer Center. Human breast cancer cells MDA-MB-231 and BT474 were obtained from Baylor College of Medicine Cell Core, whereas mouse breast carcinoma 4T1 cells were a gift from Dr. Xiang Zhang.

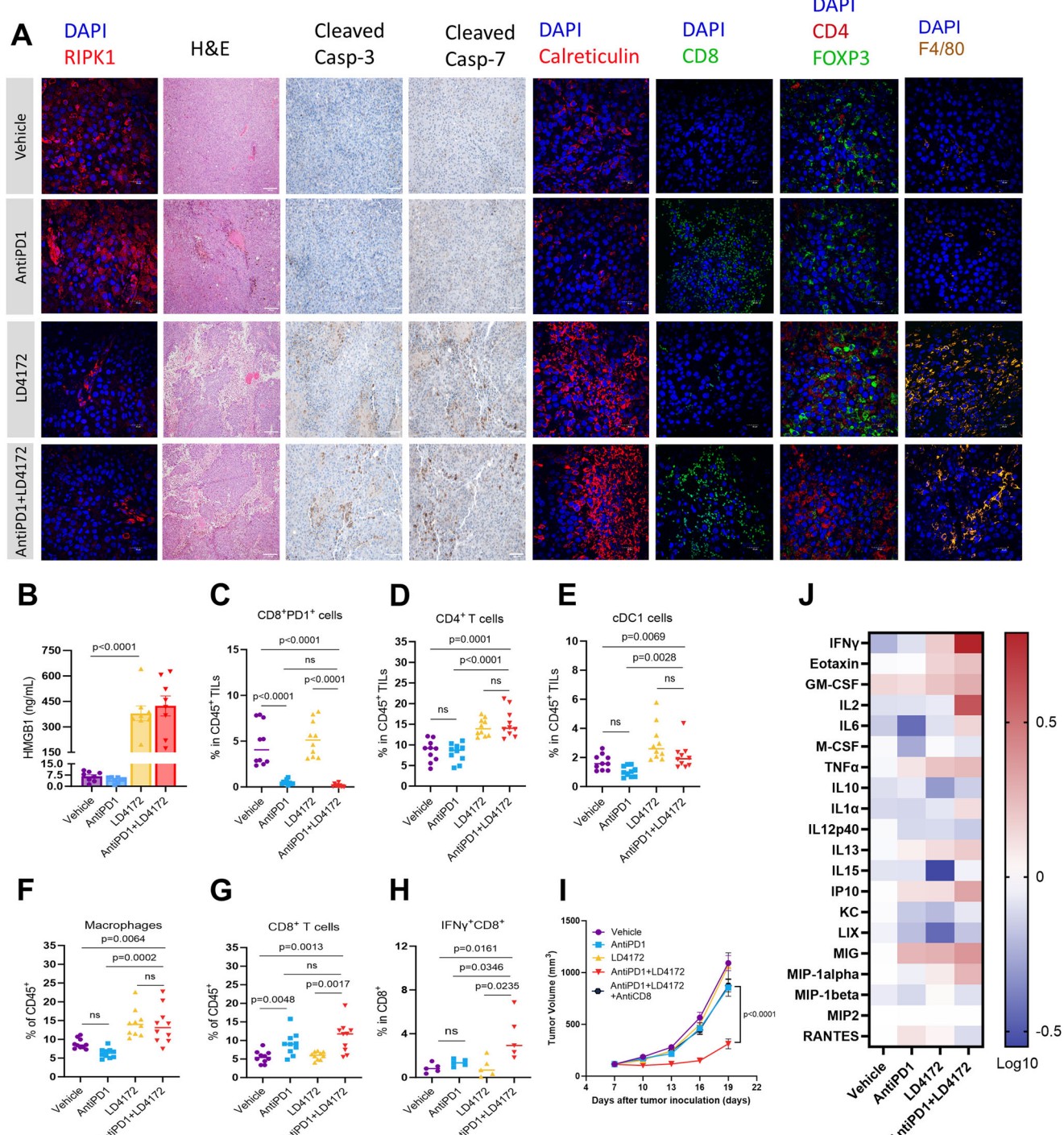

**Fig. 5 | LD4172 alters the tumor immune microenvironment. A** Representative immunofluorescent images showing RIPK1, Calreticulin, CD8, CD4, Foxp3, and F4/80 staining, along with hematoxylin and eosin (H&E) staining and cleaved caspase-3/7 levels in B16F10 tumors following 5 days of the indicated treatment. Images represent four independent fields per slide, with one slide from each of four mice per group. Scale bars: 20 μm and 200 μm (both at 60× objective). **B** The mouse plasma HMGB1 level from different treatment groups (*n* = 8 mice). Statistical significance was calculated with a two-tailed unpaired *t*-test and *P* values are indicated. Flow cytometry quantification of PD1 + CD8+ T cells **C** CD4+ T cells **D** cDC cells **E** macrophages **F** and CD8+ T cells **G** in B16F10 tumors following 5 days of indicated treatment (*n* = 10 mice/group). **H** Flow cytometric quantification of IFNγ+

CD8+ T cells in B16F10 tumors treated with the indicated treatments for 5 days and stimulated with PMA/ionomycin in vitro for 6 h (*n* = 5 tumors/group). Statistical significance was calculated with a two-tailed unpaired *t*-test and *P* values are indicated. **I** Tumor volume (mm³) of vehicle-, anti-PD1−, LD4172−, anti-PD1+LD4172−, and anti-PD1+LD4172+anti-CD8−treated B16F10 tumors (*n* = 8 mice). Each symbol represents the mean tumor volume, with error bars indicating SEM. Statistical analysis was performed using two-way ANOVA followed by Sidak's multiple comparisons test, with significance levels indicated. **J** Heat map showing log 10-fold changes in the concentration of mouse plasma cytokines normalized by the mean value of control mice (*n* = 8 mice). For all experiments: LD4172: 20 mg /kg; anti-PD1 antibody: 100 μg/mice; anti-CD8: 100 μg/mice; Combo: LD4172 plus anti-PD1.

The suspension cell lines were cultured in RPMI-1640 medium (MT10040CV, Thermo Fisher Scientific), whereas adherent cell lines were cultured in Dulbecco's modified DMEM medium (MT10013CV, Thermo Fisher Scientific). Both media were supplemented with 10% fetal bovine serum (SH30071.03, GE Healthcare), and 1% penicillin-streptomycin (15140163, Thermo Fisher Scientific). All cell lines were cultured in a humidified incubator at 37 °C with 5% $CO_2$ and routinely tested for mycoplasma contamination, with all tests confirming mycoplasma-free status.

### Induction of bone marrow-derived macrophages and dendritic cells

To isolate bone marrow cells, femur and tibia bones were dissected from 6- to 8-week-old C57BL/6 J mice. The bone marrow was flushed out into cold PBS containing 2% heat-inactivated fetal bovine serum using a 27G needle and 1 mL syringe. The cells were dissociated by passing the bone marrow through a 70 µm cell strainer and then incubated in RBC lysis buffer (1×, 420301, Biolegend) on ice for 10 min. After centrifugation (500 × $g$, 5 min, 4 °C), the supernatant was discarded, and the cells were resuspended in 1 mL of BMDM/BMDC growth medium. A suspension of $1 \times 10^6$ cells was prepared in 20 mL of cell culture medium and plated into a 10 cm petri dish. Subsequently, BMDM were induced with G-CSF (25 ng/mL, 250-05, Peprotech), and BMDCs were induced with GM-CSF (20 ng/mL, 315-03, Peprotech) and IL4 (10 ng/mL, 214-14, Peprotech).

### Western blotting

Cells were seeded into six-well plates at a density of $5 \times 10^5$ cells/mL in 2 mL of complete culture medium. Following an overnight adaptation period, cells were treated with serially diluted LD4172 compounds for 24 h. After treatment, whole-cell lysates were prepared using a lysis buffer (1×RIPA supplemented with protease and phosphatase inhibitor cocktail). Protein concentrations in the lysates were measured using the BCA protein assay. Subsequently, equal amounts of protein (20 µg) from each sample were loaded onto a sodium dodecyl sulfate-polyacrylamide gel and separated by electrophoresis (Bio-Rad) at 120 V for 1.5 h. The separated proteins were then transferred to a polyvinylidene fluoride (PVDF) membrane using a Transblot Turbo system (Bio-Rad).

After blocking for 1 h at room temperature in 1% BSA-TBST, the membranes were incubated overnight at 4 °C with specific primary antibodies (diluted at 1:1000 in TBST) targeting the proteins of interest, including anti-RIPK1 (3493, Cell Signaling Technology (CST)), anti-cleaved caspase 3 (9661, CST), anti-cleaved caspase7 (8438, CST), anti-cleaved PARP (5625, CST), anti-HMGB1 (3935, CST), anti-calreticulin (12238, CST), and anti-β-actin (4970, CST). The membranes were then incubated with horseradish peroxidase-conjugated secondary antibodies (1:1000, 7074, CST) for 1 h at room temperature. Immunoblots were imaged using ECL Prime chemiluminescent western blot detection reagent (R1100, Kindle Biosciences,) and visualized using an Imager (D1001, Kindle Biosciences). All western blots were processed and quantified using ImageJ software, and protein levels were normalized to β-actin loading controls.

### NF-κB reporter assay

One million B16F10 cells were seeded in T25 flask. The next day, cells were transfected with the NanoLuc Reporter Vector with NF-κB Response Element (pNL3.2.NF-kB-RE; Promega; Cat. No. N1111) using Lipofectamine 3000 transfection reagent (Invitrogen) at a ratio of 1:3, DNA:Lipofectamine. After 24 h, cells were collected and seeded at 2000 cells per well in triplicate in an opaque 96 well plate. 24 h later the media was discarded and replaced with fresh DMEM media and treated with indicated cytokine cocktail for 24 h. Reporter expression was read out using the Nano-Glo Luciferase Assay System (Promega) according to the manufacturer's instruction.

### Apoptosis detection using FITC-conjugated Annexin V/PI

Apoptosis quantification was conducted utilizing a FITC-conjugated Annexin V/PI assay kit (556547, BD Biosciences) and analyzed through flow cytometry. Briefly, $2 \times 10^5$ of B16F10 cells were seeded onto six-well plates and treated as specified for 72 h at 37 °C. Treated and untreated cells were harvested, washed with PBS, and resuspended in 100 µl of binding buffer. Subsequently, cells were stained with PI (50 µg/ml) and FITC-conjugated Annexin V (10 mg/ml) for 15 min at room temperature in the dark. After adding another 400 µl of binding buffer, the cells were subjected to LSR II Flow cytometer (BD Biosciences) for analysis, and flow cytometry data were processed using the FlowJo software.

### Extracellular ATP assay

To detect ATP secretion after treating B16F10 cells with specified treatments, the RealTime-Glo™ Extracellular ATP Assay (GA5010, Promega) was conducted following the manufacturer's protocol. In brief, $1 \times 10^4$ B16F10 cells were plated into each well of an opaque 96-well plate, after 72 h of treatment, 1X assay reagent was dispensed, and luminescence was recorded at regular intervals.

### TR-FRET biochemical binding assay

A time-resolved fluorescence resonance energy transfer (TR-FRET) assay was performed to evaluate the binding of the indicated compounds and RIPK1 by competition with a BODIPY-FL labeled RIPK1 tracer (Supplementary Information, **T2I-488**). The assay was performed in 20 µL assay buffer (50 mM Tris, pH7.5, 0.1% Triton X-100, 0.01% BSA, and 1 mM DTT) with 0.3 nM Tb-anti-GST (61GSTTLF, Cisbio), 2 nM GST-RIPK1 (R07-11G-10, SignalChem), 150 nM RIPK1 tracer, and serially diluted compounds (10,000 to 0.64 nM, 5-fold dilutions) in opaque 384-well plates. Unless specified otherwise, all assays were performed in triplicate. The assay mixtures were incubated at room temperature in the dark for 120 min, and the signals were collected using a BioTek Synergy H1 microplate reader to measure the fluorescence emission ratio (I520 nm/I490 nm) of each well using a 340-nm excitation filter, a 100-µs delay, and a 200-µs integration time. Raw data from the plate reader was used directly for the analysis. The curve-fitting software GraphPad Prism 10, version 10.1.2 was used to generate graphs and curves and determine $IC_{50}$ values.

### NanoBRET live-cell ternary complex assay

Human RIPK1 cDNA insert was cloned into pLenti6.2-ccdB-nLuc plasmid (87075, Addgene, a kind gift from Taipale Lab) using gateway cloning kit (11791020, Thermo) and standard protocol to obtain pLenti6.2-RIPK1-nLuc fusion vector. The day before transfection, 1 million HEK293T cells were plated in a 60 mm dish and allowed to grow overnight in DMEM/10% FBS. The next day, the cells were co-transfected overnight at 37 °C with 1 ng/ml pLenti6.2-RIPK1-nLuc fusion vector, 100 ng/ml HaloTag®-VHL Fusion Vector (N273A, Promega) along with 1ug/ml carrier DNA vector (E4881, Promega) using the calcium phosphate method. After 18 h, transfected cells were trypsinized and resuspended in Opti-MEM (11058-021, Gibco) supplied with 4% FBS and 100 nM HaloTag® NanoBRET™ 618 Ligand (G9801, Promega) to a cell-density of 0.2 M/ml (for background subtraction group, 618 ligand was omitted). Plate 100ul cells into each 96-well (136101, Thermo). The plate was further incubated at 37 °C overnight to allow HaloTag-VHL to be labeled with 618 ligand. Next day, cells were further treated with 10uM MG132 for 0.5 h followed by 1 µM PROTAC or DMSO for 4 h. Immediately before reading the plate, prepare 4× concentrated NanoGlo nLuc substrate (N157, Promega) was prepared by diluting the stock into Opti-MEM 1000-fold, and the NanoGlo substrate was then added into each well to bring the final concentration to 1×. Donor emission at 450 nM and acceptor emission at 610 nM were measured on a BioTek Synergy H1 plate reader equipped with filter cube set 450/80 and 610 LP. The corrected mBU

was calculated as follows:

$$\text{Corrected mBU} = \left\{ \left[ \frac{Em_{618nm}}{Em_{450nm}} \right]_{\text{PROTAC or DMSO}} - \left[ \frac{Em_{618nm}}{Em_{450nm}} \right]_{\text{without 618 ligand}} \right\} \times 1{,}000 \quad (1)$$

## Proteomics study

One million MDA-MB-231 cells were seeded in 6-well plates. The following day, cells were treated in triplicate with LD4172 (200 nM) or LD4172-NC (200 nM) for 6 h. The cells were washed thrice with ice-cold PBS. The cell pellets were lysed, reduced, alkylated, and digested using EasyPep™ MS Sample Prep Kits (A45733, ThermoFisher) according to the manufacturer's instructions. The same amount of peptide from each condition was labeled with a tandem mass tag (TMT) reagent (90113, ThermoFisher). The 10-plex TMT reagent was incubated with each peptide sample at a ratio of 1:8 (peptide:TMT label). The 10-plex labeling reactions were performed for 1 h at room temperature. The labeled peptide samples were quenched by adding 50 μL of 5% hydroxylamine and 20% formic acid solution for 5 min and then mixed. The mixed samples were desalted and fractionated offline into 24 fractions on a 250 × 4.6 mm Zorbax 300 Extend-C18 column (Agilent) using an Agilent 1260 Infinity HPLC system.

The 24 fractions were dried in vacuo and resuspended in 5% acetonitrile in water (0.1% FA). Each sample was first separated by nano LC through a 5–40% ACN gradient within 75 min and ionized by electrospray (2.4 kV), followed by MS/MS analysis using higher-energy collisional dissociation (HCD) at a fixed 38.0 collision energy on an Orbitrap Fusion Lumos mass spectrometer (Thermo Fisher Scientific) in data-dependent mode with a 3 s cycle-time. MS1 data were acquired using the FTMS analyzer in profile mode at a resolution of 120,000 over a range of 400–1600 m/z. Following HCD activation and quadrupole isolation with a window of 0.7 m/z, MS/MS data were acquired using an orbitrap at a resolution of 50,000 in centroid mode and normal mass range.

Proteome Discoverer 2.4 (Thermo Fisher Scientific) was used. RAW file processing and controlling peptide- and protein-level false discovery rates, assembling proteins from peptides, and protein quantification from peptides. Searches were performed using full tryptic digestion against the SwissProt human database with up to two miscleavage sites. Oxidation (+15.9949 Da) of methionine and Deamidation on N and Q (0.984 Da) were set as variable modifications, while carbamidomethylation (+57.0214 Da) of cysteine residues and TMT 10-plex labeling of peptide N-termini and lysine residues were set as fixed modifications (+229.163 Da). Data were searched with mass tolerances of ±10 ppm and 0.02 Da on the precursor and fragment ions (HCD), respectively. The results were filtered to include peptide spectrum matches (PSMs) with a high peptide confidence. PSMs with precursor isolation interference values >50% and average TMT-reporter ion signal-to-noise values (S/N) < 10 were excluded from quantitation. Isotopic impurity correction and TMT channel normalization, based on the total peptide amount, were applied. Protein quantification uses both unique and random peptides. For statistical analysis and adjusted $p$-value calculation, an integrated analysis of variance (ANOVA) hypothesis test on individual proteins was used. TMT ratios with adjusted $p$-values below 0.01 were considered significant.

The MS raw data files for quantitative multiplexed proteomics have been deposited in the MassIVE dataset under accession number MSV000092377.

## Molecular docking

Molecular docking studies were carried out using Schrödinger software. Schrödinger adopted the Glide algorithm to dock flexible ligands into the protein-binding site. The crystal structure of the RIPK1 kinase domain in complex with the isoquinolin-1amine analog (PDB: 4NEU) was used as the receptor structure in molecular docking studies.

## Generation of CRISPR-edited tumor cell lines

RIPK1 genetic deletion was accomplished using the Neon transfection system (ThermoFisher). Five nanometer of sg-RNA targeting mouse RIPK1 (sgRIPK1 #1: GGGTCTTTAGCACGTGCATC, sgRIPK1 #2: CAGTC-GAGTGGTGAAGCTAC) or non-targeting negative control sgRNA (sgNC #1: GAAGATGGGCGGGAGTCTTC) was mixed with 2 nM Cas9 enzyme (IDT) at room temperature for 15 min to generate the RNP complex, which was then electroporated into $4 \times 10^5$ B16F10 cells. The medium was replaced 24 h after electroporation. Single-cell clones were screened for protein expression by western blotting. Confirmed gene-deleted clones were pooled and cultured for 2 weeks in vitro before being implanted in vivo.

## Animal studies

All animal experiments were conducted in accordance with the protocol approved by the Institutional Animal Care and Use Committee of Baylor College of Medicine. Tumor studies were performed exclusively in female mice. Six-week-old female C57BL/6J mice, ordered from Jackson Labs, were used for experiments and age-matched for consistency. Female mice were chosen for their balanced representation and stable exploratory behavior. Mice were housed in the TMF Mouse Facility at Baylor College of Medicine under SPF conditions with climate control and 12-h light/dark cycles. Fresh chow and water were provided continuously through an automated water system. The investigators were blinded to group allocation during data collection and/or analysis.

## Animal treatment and tumor challenges

To establish a syngeneic mouse model, $3 \times 10^5$ B16F10 cells resuspended in PBS were mixed 1:1 with Matrigel (354262, Corning) and subcutaneously injected into the right flank of 7-week-old wild-type female C57BL/6J mice on day 0. Tumor sizes were measured on day 6, and mice were grouped such that the average tumor volume was ~100 mm³. Treatments began on day 7. Antibodies, including anti-PD1 (100 μg, BE0146, BioXcell) and anti-CD8 (500 μg, BE0004, BioXcell), were administered intraperitoneally every 3 days in 100 μL volumes. The RIPK1 degrader LD4172 and RIPK1 kinase inhibitor T2I were delivered in 200 μL of vehicle solution (30% PEG400, 5% Tween-80, 5% DMSO) daily via intraperitoneal injection (20 mg/kg). T-cell depletion or blocking was confirmed by flow cytometry in tumor-bearing mice. Tumors were measured every 3 days starting from day 7 post-injection until euthanasia. Mice were euthanized via $CO_2$ inhalation when the tumor size reached 1.5 cm in the longest dimension or if severe necrosis was observed. Tumor volume was calculated using the formula: $(L \times W^2)/2$, where $L$ represents the length and $W$ the width. To comply with ethical guidelines and ensure data quality, animals were humanely euthanized either when tumors reached the maximum permitted size or immediately before flow cytometry analysis, which required fresh tumor tissues. In cases where tumors exceeded the maximum allowable volume (1700 mm³), this was attributed to the unexpected aggressive growth of B16F10 tumors in the later stages of development.

## Analysis of tumor-infiltrating lymphocytes (TIL) by flow cytometry

Tumors were collected on day 13, weighed, mechanically diced, and digested with liberase (2 mg/mL, 05401020001, Roche) and DNase I (50 μg/mL, 11284932001, Sigma-Aldrich) at 37 °C for 30 min with rotation. Single-cell suspensions were obtained by filtering the digested tissues through a 45 μm strainer, after which erythrocytes were removed using 1× RBC lysis buffer (420301, Biolegend). To stain the cell surface markers of TILs, single-cell suspensions were blocked with anti-mouse CD16/32 (156603, BioLegend) for 10 min on ice, and then incubated with fluorochrome-labeled antibodies diluted with staining buffer (1:50, PBS, 2% FBS, 0.1% EDTA) for 30 min on ice in the dark. Dead cells were excluded using DAPI (1:1000, BioLegend). After

washing, cells were resuspended in 300–500 μL staining buffer for flow cytometry analysis. For intracellular cytokine staining, cells were incubated in culture medium (RPMI-1640, 10% FBS, Brefeldin A, or Cell Activation Cocktail (with Brefeldin A)) at 37 °C for 6 h. Cells were stained with eBioscience™ Fixable Viability Dye eFluor™ 450, TruStain FcX™ PLUS and surface CD45, CD3ε, CD8a. Cells were fixed-permeabilized by Intracellular Fixation & Permeabilization Buffer Set (Biolegend, 421403) according to the manufacturer's protocol. Then cells were stained with anti-IFNγ (XMG1.2) for 1 h at room temperature (RT). Cells were resuspended in FACS buffer and subjected to BD LSR-II for analysis.

Lymphoid cell phenotyping panel: CD45-APC750 (30-F11), CD3e-APC (145-2C11), CD4-BV650 (GK1.5), CD8-PercpCy5.5 (53-6.7), PD1-PE (RMP1-14), DAPI. Myeloid cell phenotyping panel: CD45-APC750, I-A/I-E-APC (M5/114.15.2), CD11c-PE (N418), CD11b-PercpCy5.5 (M1/70), Ly6C-AF700 (HK1.4), F4/80-FITC (BM8), XCR1-BV650 (ZET).

All data were acquired using LSRII Analyzer and analyzed with Flow Jo v10.0.

## Cytokine array and enzyme-linked immunosorbent assay (ELISA)

Mouse serum was analyzed using multiplex immunoassays designed for mice (Mouse Cytokine Array/Chemokine Array 31-Plex (MD31) from Eve Technologies), with 8 replicates from each group. Heatmaps display relative cytokine expression values normalized to vehicle-treated samples. Serum HMGB1 was analyzed by ELISA (NOVUS, NBP2-62767) according to the manufacturer's protocol.

## Immunohistochemistry and immunofluorescence

The mouse tumors were fixed in 4% paraformaldehyde overnight at 4 °C, washed, and then stored in 70% ethanol until paraffin embedding. Paraffin sections (5 μm) were hydrated for subsequent analysis.

For H&E staining, the hydrated slides were stained with hematoxylin and eosin. For IHC analysis, after hydration, the sections were subjected to antigen retrieval by incubating in citrate buffer (pH 6.0), Tris EDTA buffer (pH 9.0), or EDTA buffer (pH 8.0) at 121 °C for 15 min. Endogenous peroxidase was blocked with 3% $H_2O_2$ in PBS, and non-specific binding was blocked with 2.5% normal serum for 1 h at room temperature. The sections were then incubated with the respective primary antibodies overnight at 4 °C. The primary antibodies used in IHC were cleaved caspase 3 (1:200, 9661, CST) and cleaved caspase 7 (1:200, 8438, CST). Following PBS washes, the sections were incubated with secondary antibodies (SignalStain Boost IHC Detection Reagent, 8114, CST), developed with DAB (SignalStain DAB Substrate,8059, CST), and counterstained with hematoxylin (VWR, 100504-658). At least four representative images of tumor sections from each group were acquired.

For immunofluorescence analysis, slides after antigen retrieval were incubated overnight at 4 °C with the following primary antibodies: CD8 (1:100, 14-0808-82, eBioscience), CD4 (1:100, NBP1-19371, Novus Biologicals), FOXP3 (1:100, NB100-39002, Novus Biologicals), or F4/80 (1:100, NB600-404, Novus Biologicals). After washing with PBS, the tumor sections were incubated with Alexa Fluor 488/594/647 secondary antibodies (1:500; Thermo Fisher Scientific) for 1 h at room temperature, followed by nuclei staining with DAPI for 20 min (1:30,000, 422801, Biolegend). The slides were mounted with Pro-Long™ Diamond Antifade Mountant (Thermo Fisher, P36970) and imaged using a Zeiss LSM780 confocal microscope with a ×60 objective. Consistent image exposure times and threshold settings were applied for all groups.

## Statistical information

Data are presented as mean ± SD (standard deviation) or ±SEM (standard error of the mean), calculated using GraphPad Prism 10, version 10.1.2. A $P$ value of <0.05 was considered statistically significant. Specific $P$ values and the statistical tests used are provided within each figure legend. Statistical tests include unpaired two-tailed Student's $t$-tests or two-way ANOVA followed by Sidak's multiple comparisons test, as indicated. $N$ values in the figure legends represent biological replicates, while technical replicates refer to repeated measurements from the same samples. Statistical analyses are based on averaged values across biological replicates, not pooled technical and biological replicates.

## Reporting summary

Further information on research design is available in the Nature Portfolio Reporting Summary linked to this article.

## Data availability

The mass spectrometry raw data files for quantitative multiplexed proteomics have been deposited in the MassIVE dataset under accession number MSV000092377 (see also: https://proteomecentral.proteomexchange.org/cgi/GetDataset?ID=PXD043614). The remaining data are available within the Article, Supplementary Information or Source Data file. Source data are provided with this paper and are also available in Figshare [https://doi.org/10.6084/m9.figshare.27629700] Source data are provided with this paper.

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

## Acknowledgements

This research was supported by the National Institutes of Health (R01-CA250503 and R01-CA268518 to J.W.), the Cancer Prevention & Research Institute of Texas (CPRIT, RP220480 to J.W.), and the Howard Hughes Medical Institute (to M.C.W.). Additional support was provided by Michael E. DeBakey, M.D., Professorship in Pharmacology (to J.W.). We are grateful to Drs. Ying Li, Douglas Green, Pascal Meier, Jeff Rosen, and Xiang Zhang for their valuable suggestions and advice. We also acknowledge the assistance of the Cytometry and Cell Sorting Core at Baylor College of Medicine, which is funded by the CPRIT Core Facility Support Award (CPRIT-RP240432) and the NIH (CA125123 and OD036336), with special thanks to Joel M. Sederstrom for his technical support.

## Author contributions

X.Y. designed and performed the experiments, analyzed and interpreted the data, and wrote the manuscript. D.L. designed and synthesized the RIPK1 degraders and contributed to manuscript writing. X.Q. established the xenograft tumor model and managed protocol logistics. B.L.H. assisted with animal studies. R.R.P., L.D., and F.J. conducted mass spectrometry for PK studies. D.L., F.J., and H.F.L. developed methods for the proteomic study and performed data analysis. L.Y.X., W.Y.P., X.C., and M.C.W. provided project supervision, contributed to data interpretation, and substantially revised the manuscript. J.W. supervised the entire study, managed protocol logistics, interpreted data, wrote the manuscript, and secured funding.

## Competing interests

J.W. is a co-founder of Chemical Biology Probes, LLC, and serves as a consultant for CoRegen Inc. X.Y., D.L., and J.W. are inventors on a patent covering RIPK1 degraders reported in this work, titled "Novel RIPK1 Kinase-Targeting PROTACs and Methods of Use Thereof", with the identification number WO2022120118A1. The remaining authors declare no competing interests.
