## [Peer Review file · Nature Communications]

Development of a RIPK1 Degradator to Enhance Antitumor Immunity

Corresponding Author: Professor Jin Wang

Version 0:

Reviewer comments:

Reviewer #1

(Remarks to the Author)

This manuscript by Yu et al describes the development and testing of a novel RIPK1 protac LD4172. Using a RIPK1 type II kinase inhibitor attached to a VHL ligand, the authors showed that LD4172 can effectively degrade RIPK1 in multiple human and mouse tumor cell lines in vitro, as well as in some but not all tissues in vivo. The compound also sensitized B16F10 cells to apoptosis induced by TNF in vitro. This is in line with reports in the literature that RIPK1 has a pro-death function that is dependent on its kinase, as well as a pro-survival function via its intermediate and death domain. The latter scaffold function has been suggested to be due to RIPK1's role in stabilizing pro-survival molecules such as TRAF2, and/or its role in activating NFkappaB. RIPK1 also has a pro-survival role by blocking the interaction between ZBP1 and RIPK3 to suppress RIPK3-dependent death. In vivo, LD4172 enhanced the sensitivity of B16 F10 tumors to immune checkpoint blockade by anti-PD1. The authors suggests that tumor death conferred by RIPK1 degradation is immunogenic based on HMGB1 release, calreticulin exposure, and recruitment of various inflammatory cells to the dying tumor microenvironment. Quite interestingly, RIPK1 degradation seems to be much more pronounced in the tumor tissue than in non-tumor tissues. Authors speculated that this could be due to the compound binding to albumin, which may accumulate in tumor sites. This is largely a descriptive study of the effect of administrating the RIPK1 protac in vitro and in vivo. Overall, this is a thorough study that provides proof-of-concept for targeting RIPK1 in a preclinical setting.

1. The main deficiency is that there is not much mechanistic dissection. The prior publication from Cucolo et al (Ref #5) had suggested that in the absence of RIPK1, there is a shift from Complex I to Complex II in the TNFR1 pathway. It would be nice to show that this is also true when RIPK1 is acutely removed by the protac.
2. While the in vitro studies showed that LD4172 sensitizes to TNFR1-induced apoptosis, it is less clear whether this is the case in vivo or whether other death receptors in the TNFRSF family may also be involved. If TNF/TNFR1 signaling is responsible for the sensitizing effect of LD4172 in vivo, then one would predict that B16 F10 cells that are knockout for TNFR1 would be resistant to the effect of LD4172 + anti-PD1. Experiment in Fig 4F can be repeated comparing WT vs TNFR1 KO B16 cells. It would be interesting to know one way or the other.
3. While the proteomic analysis in Fig. 2J indicates a high degree of specificity, an additional control that could be included is to show that the related kinases RIPK2 and RIPK3 are not affected by LD4172.
4. In the legend for Figure 3, the early apoptotic cells (AV+, PI-) should be in the upper left quadrant, not lower right.
5. The key to the graph in Fig S2A for the MC38 experiments appears to be incorrect. Shouldn't it be PD1 and LD4172 treatment instead of RIPK1 KO?

Reviewer #2

(Remarks to the Author)

The authors have generated a PROTAC targeting RIPK1 for degradation and showed that it recapitulates the effect of loss of RIPK1 in tumours in vitro and in vivo. Interestingly, their RIPK1 PROTAC synergises with anti-PD1 therapy in vivo. This study is generally well designed and controlled. To my knowledge, it is the first RIPK1 PROTAC showing such good synergism in vivo. Given the lethality of RIPK1 mice, this tool can also be used to study the role of RIPK1 in vivo in adult development, infectious diseases, etc. There are a few imprecisions and clumsiness throughout the manuscript that need to be addressed. Please see specific comments.

Specific comments:

In Fig. 2D-E, the authors should acknowledge that RIPK1 is already re-expressed after a 4-hour washout. This could be important in terms of RIPK1-dependent responses, as we don't know what level of RIPK1 is required to fully reconstitute its function. For example, there is a small proportion of RIPK1 that goes to TNF-induced Complex I; therefore, 10% of RIPK1 (after 4-hour washout) could be enough to provide a normal TNF response.

Line 125 'In contrast to situations where RIPK1 is kinase-dead, genetic deletion of RIPK1 has been found to trigger apoptosis both in vitro and in vivo 15.'

Ref 15 is only about RIPK1 knock-out mice and not about RIPK1 kinase dead. Plus, ref 15 is not the only one and the first one to show that loss of RIPK1 induces apoptosis. The authors should be fair and cite all RIPK1 kinase dead and RIPK1 knock-out papers that support their statement, e.g., Kelliher Immunity 1998; Kaiser PNAS 2014; Rickard Cell 2014; Polykratis JI 2014; Berger JI 2014; Newton Science 2014.

In Fig S2A the legend says that it's sgNC vs RIPK1 KO, while in the manuscript, the authors described it as tumours being treated with LD4172. Is it a mistake with the legend or the wrong graph?

In Line 181 the authors wrote "Consistent with the in vitro findings, LD4172 also triggered significant cell death in the tumor (Fig. 182 5A, 2nd column).

The 2nd column is H&E, which does not reflect of cell death but rather absence or presence of cells. Caspase-3/7 are the cell death markers. Please correct the text accordingly.

The authors wrote in line 184 'While apoptotic cell death was traditionally considered non-immunogenic, accumulating experimental data have revealed its potential to drive immune cell infiltration and anti-cancer immunity 16–19. Supporting the activation of immunogenic apoptosis, we observed a significant increase in plasma HMGB1 levels (Fig. 5B) and enhanced exposure of calreticulin on the surface of B16F10 tumor cells (Fig. 5A, 5th column).'

However, the authors don't have supporting evidence that apoptosis is the mode of cell death that is immunogenic because it is well established that loss of RIPK1 can also induce RIPK3-dependent necroptosis (Kaiser PNAS 2014; Rickard Cell 2014; Dillon cell 2014; Berger JI 2014 ; Newton Science 2016), which is also immunogenic. One way to determine if apoptosis or necroptosis is immunogenic is to analyse MLKL^{-/-} and casp3/7^{-/-} tumours treated with RIPK1 PROTAC. The authors should rewrite this part.

The authors wrote in line 196 'In addition, combined therapy with LD4172 and anti-PD1 not only induced extensive TIL infiltration (Fig. 5D-H) but also significantly enhanced anti-PD1 positivity in immunologically cold B16F10 tumors, as demonstrated by increased infiltration of cytotoxic CD8⁺ T cells (CD8⁺IFN- γ ⁺, Fig. 5A, 6th column, and 5G-H) and decreased infiltration of FOXP3⁺ T regulatory cells (Fig. 5A, 7th column) within the TME.'

This is wrong: LD4172 did not increased TIL infiltration CD8⁺ T cells induced by anti-PD1 because there is not statistical difference between anti-PD1 treatment and anti-PD1+ LD4172 treatment in Fig 5C, G and H. Please correct this part.

The authors wrote in line 235 'Unlike Ripk1 knockout mice, which die at 1-3 days of age due to their widespread roles in multiple tissues and organs 25, homozygous loss-of-function RIPK1 mutations are well tolerated in humans 26. Patients with complete loss of RIPK1 protein only showed symptoms confined to the immune system, with primary immunodeficiency and/or intestinal inflammation 26.'

And Line 240 'Although the safety profiles of RIPK1 degraders remain to be tested in future clinical studies, human genetic data suggest that pharmacological RIPK1 degradation is potentially safe and tolerable, especially with transient intervention in well-controlled clinical settings.'

And Line 259 'Considering the predicted safety profile of RIPK1 degradation based on human genetics'

By stating that loss of RIPK1 is 'well tolerated in humans', 'only showed symptoms confined to the immune system', 'human genetic data suggest that pharmacological RIPK1 degradation is potentially safe and tolerable' and 'predicted safety profile of RIPK1 degradation based on human genetics', the authors are minimising the effect of loss of RIPK1 in human to promote the use of RIPK1 PROTAC. This is extremely clumsy because Ref 26 and Li et al PNAS 2019

(doi.org/10.1073/pnas.18135821, which should be cited alongside Ref 26) showed that RIPK1 deficient patients are affected with severe immunodeficiency which can lead to the death of the patients at really young age. The authors should rephrase this part by stating that despite the severe immunodeficiency cause by permanent loss of RIPK1 in human, acute, chemical and transient depletion of RIPK1 might be tolerable in humans. Although the safety profiles of RIPK1 degraders remain to be tested in future clinical studies.

Reviewer #3

(Remarks to the Author)

The authors reported the development of the first-in-class RIPK1 degraders that utilize the VHL E3 ligase to potently degrade RIPK1 in various cells. One of the lead compounds, LD4172, is highly specific and only degrade RIPK1 in cells with high Dmax and DC50 values in a VHL and proteasome-dependent manner. LD4172 can synergistically kill B16F10 mouse melanoma cells in combination with TNFa. Interestingly, administration of B16F10 tumor-bearing mice with LD4172 resulted in significant reduction in RIPK1 expression in the tumors but not in normal tissues. This reduction was associated with the potentiation of tumor growth inhibition when it was combined with anti-PD1 antibody. More importantly, the synergistic effect was greater with the combination of anti-PD1 antibody and LD4172 than that of anti-PD1 antibody and the

RIPK1 inhibitor. These findings demonstrate that RIPK1 degraders may exert stronger antitumor activity than RIPK1 inhibitors by degrading RIPK1 to remove both its kinase and scaffold activity. In addition, this combination can also enhance antitumor immunity.

Concerns:

1. It is very interesting that only VHL-based PROTACs can effectively degrade RIPK1 but not CRBN- and MDM2-based and adamantane-tagged PROTACs. Can the authors share some insights why only VHL can degrade RIPK1? Have the authors tried to use different linker length for other PROTACs to validate the lack of degradation for RIPK1?
2. The authors showed that LD4172 can sensitize B16F10 cells to TNF α . Since they also showed that LD4172 can synergistically suppress B16F10 tumor growth with Anti-PD1 antibody, have the authors tested whether LD4172 can sensitize B16F10 cells to cytotoxic T cells?
3. The finding that LD4172 was more effectively in degrading RIPK1 in tumor tissues than in normal tissues is very interesting. Do the authors explore why these normal tissues exhibited less RIPK1 degradation after the treatment with LD4172? Is this related to different uptake of the compounds between tumors and normal tissues?
4. RIPK1 KO has a profound effect in mice as they die at 1-3 days of age while patients with homozygous loss-of-function RIPK1 are alive and only exhibit some less severe abnormalities such as primary immunodeficiency and/or intestinal inflammation. However, the mice treated with LD4172 did not show any side effects in mice. This may be attributable to the inability of LD4172 to effectively degrade RIPK1 in normal tissues. This finding may have important implications in developing RIPK1 targeted therapy because RIPK1 might be more toxic to normal tissues while RIPK1 PROTACs might be more selective to tumors. As such, RIPK1 PROTACs may have a better opportunity to be developed as anticancer drugs.
5. The finding that depletion of CD8 T cells with an antibody abrogated the synergistical antitumor activity of LD4172 and anti-PD1 antibody suggests that the direct antitumor activity of RIPK1 PROTACs is less important than the induction of CD8 T cell-mediated antitumor immunity in the B16F10 tumor model.

Reviewer #4

(Remarks to the Author)

I support publication of this work after addressing comments below.

- What are the noteworthy results?

This is a nice study describing the discovery and characterization of the first PROTAC-based RIPK1 degrader. The study appears to be done with excellent quality and the conclusions are well supported by the provided data.

Key results:

1. RIPK1 can be degraded using a suitably designed/optimized PROTAC degrader
2. RIPK1 degrader has fundamentally different pharmacology relative to RIPK1 inhibitor
3. Degradation of RIPK1 by LD4172 triggered immunogenic cell death (ICD)
4. Degradation of RIPK1 resulted in an increase in tumor-infiltrating lymphocytes and sensitized tumors to anti-PD1 therapy

The efficacy in the mouse models is limited (this would be called progressive disease in patients) but does show the mechanism appears to be active in a murine model.

It would be nice to see:

1. PK data for LD4172 to enable understanding of how much drug exposure there was following the dosing regime used.
 2. To show evidence of target degradation in vivo (blood and tumor tissue).
- Will the work be of significance to the field and related fields? How does it compare to the established literature? If the work is not original, please provide relevant references.

Yes demonstrates another use of small molecule degraders to achieve novel pharmacology.

- Does the work support the conclusions and claims, or is additional evidence needed?

The work is well supported by the data provided.

- Are there any flaws in the data analysis, interpretation and conclusions? Do these prohibit publication or require revision?

None that I could find.

- Is the methodology sound? Does the work meet the expected standards in your field?

- Is there enough detail provided in the methods for the work to be reproduced?

Yes experimental sections appear to be sound, nice characterization data is provided for the compounds.

Version 1:

Reviewer comments:

Reviewer #1

(Remarks to the Author)

This revised manuscript by Yu et al has addressed the critiques of this reviewer. While an experiment using TNFR1 KO of the B16 F10 cells would have been preferable in the in vivo studies, the authors used anti-TNF to show that the anti-tumor effect of the LD4172 protac is TNF-dependent. A recent published study from Pascal Meier's group also showed similar findings with a different RIPK1 protac and in that study, some of the effects of a RIPK1 protac on TNFR1 signaling complexes were examined. Some of the sloppiness in the manuscript brought up by this and other reviewers were also addressed. The manuscript is now acceptable.

Reviewer #2

(Remarks to the Author)

The authors have adequately addressed all my comments.

Reviewer #3

(Remarks to the Author)

The revised manuscript has significantly improved with new experimental data to address all my concerns. Particularly, the data from the LD4172 tissue distribution study and RIPK1 degradation in tumors vs various normal tissues are very interesting. I agree with the authors that albumin binding and higher uptake and degradation of albumin by tumor cells may contribute to the differences between tumors and normal tissues observed. It would be interesting to explore this further in future studies.

Dear Editor and Reviewers:

We have completed the revisions for our manuscript titled “Development of a First-in-
Class RIPK1 Degradator to Enhance Antitumor Immunity”. We sincerely appreciate the
thorough and professional review of our work. As noted, there were several issues that
needed to be addressed. In response to the reviewers’ suggestions, we have made
significant corrections to the manuscript, which are highlighted in the revised version.
Our detailed responses to the reviewers' comments are provided below.

Thank you for your continued consideration.

**Detailed responses to the reviewers' comments:**

**Reviewer #1** (Remarks to the Author): with expertise in cancer immunotherapy, cell
death signaling, RIPK1

This manuscript by Yu et al describes the development and testing of a novel RIPK1
protac LD4172. Using a RIPK1 type II kinase inhibitor attached to a VHL ligand, the
authors showed that LD4172 can effectively degrade RIPK1 in multiple human and
mouse tumor cell lines in vitro, as well as in some but not all tissues in vivo. The
compound also sensitized B16F10 cells to apoptosis induced by TNF in vitro. This is in
line with reports in the literature that RIPK1 has a pro-death function that is dependent
on its kinase, as well as a pro-survival function via its intermediate and death domain.
The latter scaffold function has been suggested to be due to RIPK1’s role in stabilizing
pro-survival molecules such as TRAF2, and/or its role in activating NFkappaB. RIPK1
also has a pro-survival role by blocking the interaction between ZBP1 and RIPK3 to
suppress RIPK3-dependent death. In vivo, LD4172 enhanced the sensitivity of B16 F10
tumors to immune checkpoint blockade by anti-PD1. The authors suggests that tumor
death conferred by RIPK1 degradation is immunogenic based on HMGB1 release,
calreticulin exposure, and recruitment of various inflammatory cells to the dying tumor
microenvironment. Quite interestingly, RIPK1 degradation seems to be much more
pronounced in the tumor tissue than in non-tumor tissues. Authors speculated that this
could be due to the compound binding to albumin, which may accumulate in tumor
sites. This is largely a descriptive study of the effect of administrating the RIPK1 protac
in vitro and in vivo. Overall, this is a thorough study that provides proof-of-concept for
targeting RIPK1 in a preclinical setting.

1. The main deficiency is that there is not much mechanistic dissection. The prior
publication from Cucolo et al (Ref #5) had suggested that in the absence of RIPK1,
there is a shift from Complex I to Complex II in the TNFR1 pathway. It would be nice to
show that this is also true when RIPK1 is acutely removed by the protac.

Response: Thank you for your insightful comment. You are absolutely correct in noting
that RIPK1, known for its scaffold function, is recruited into complex I upon TNF- α
signaling. Following its ubiquitylation, the recruitment of the TAK1-TAB2/3 complex and
the NEMO-IKK α -IKK β complex is crucial for the activation of NF- κ B signaling. Our NF-
κ B reporter assay results support this, showing significant NF- κ B activation by TNF- α in
B16F10 cells (Figure 3A). As expected, the RIPK1 degrader LD4172, as opposed to the
RIPK1 kinase inhibitor T2I, significantly inhibited TNF- α -induced NF- κ B activation.
Furthermore, the combination of TNF- α with LD4172 led to a marked induction of
apoptosis in B16F10 cells (Figure 3B-D), indicating that acute deletion of RIPK1 shifts
the cells from a pro-survival to a pro-death state. Interestingly, a recent study by the
Meier group, published in Immunity, reported the development of a RIPK1 PROTAC
degrader, R1-ICR-5. Their findings demonstrate that RIPK1 degradation can promote
the interactions of TNFR1, TRADD, TRAF2, cIAP, and HOIP, forming a RIPK1-
independent complex I, with enhanced ubiquitylation leading to deregulated TNF
signaling (Figure adapted from Mannion's paper). These findings underscore the
potential of RIPK1 degraders as powerful chemical probes for investigating the
biological roles of RIPK1 in complex I and complex II. We have incorporated these
updates in lines 130-137,227-237 and Figure 3A.

[figure redacted]

Adapted from Mannion's paper: Acute degradation of RIPK1 deregulates TNFR1
signaling, L929 Cells were treated with DMSO or R1-ICR-5 (overnight) before anti-GST-
TUBE pull-down to isolate the ubiquitylated proteome.

Figure 3A. NF- κ B activity of B16F10 cells expressing a NanoLuc reporter for NF- κ B
response with indicated treatments for 48 hours.

Updated paragraph in the manuscript can be found as follows:

Lines 130-137: "RIPK1-deficient MEFs exhibit a severe impairment in their ability to
activate NF- κ B signaling, whereas kinase-dead RIPK1 knock-in mice remain viable and
display normal TNFR1-mediated NF- κ B signaling¹⁶⁻¹⁸. Moreover, unlike the kinase-dead
RIPK1 scenario, the genetic deletion of RIPK1 has been shown to trigger apoptosis
both *in vitro* and *in vivo*¹⁹⁻²⁴. To evaluate NF- κ B activity in B16F10 cells, we transiently
transfected them with a plasmid encoding a Nanoluc reporter containing the NF- κ B
response element, allowing us to monitor NF- κ B activity through luminescence. Upon
TNF- α treatment, B16F10 cells exhibited a robust induction of NF- κ B activity, which was
significantly attenuated by LD4172 but not by the RIPK1 kinase inhibitor (Fig. 3A)."

Lines 227-237: "Upon TNF- α signaling, RIPK1 is recruited to complex I, where it acts as
a crucial scaffold, essential for the ubiquitylation-dependent activation of NF- κ B
signaling. Our findings demonstrate that LD4172-induced RIPK1 degradation, unlike the
action of RIPK1 kinase inhibitors, significantly impairs TNF- α -induced NF- κ B activation.
Furthermore, the combination of TNF- α and LD4172 markedly induces apoptosis in
B16F10 cells. This suggests that acute RIPK1 deletion shifts the cellular response from
a pro-survival to a pro-death state by altering the composition of complexes I and II.
Interestingly, the Meier group developed a similar RIPK1 degrader¹⁸, revealing that
RIPK1 degradation promotes the formation of a RIPK1-independent complex I. This
complex display enhanced interactions among TNFR1, TRADD, TRAF2, cIAP, and
HOIP, leading to deregulated TNF signaling through increased ubiquitylation. These
findings underscore the potential of RIPK1 degraders as valuable chemical tools for *in*
*vitro* studies, offering new insights into the biological roles of RIPK1 within various
signaling complexes."

2. While the *in vitro* studies showed that LD4172 sensitizes to TNFR1-induced
apoptosis, it is less clear whether this is the case *in vivo* or whether other death
receptors in the TNFRSF family may also be involved. If TNF/TNFR1 signaling is
responsible for the sensitizing effect of LD4172 *in vivo*, then one would predict that B16
F10 cells that are knockout for TNFR1 would be resistant to the effect of LD4172 + anti-
PD1. Experiment in Fig 4F can be repeated comparing WT vs TNFR1 KO B16 cells. It
would be interesting to know one way or the other.

Response: Thank you for highlighting this important aspect that we initially overlooked
in our study of LD4172. As you suggested, the sensitizing effect of LD4172 *in vivo* is
indeed dependent on TNF/TNFR1 signaling. Our findings demonstrate that neutralizing
TNF α with anti-mouse TNF α *in vivo* completely abolishes the synergy between LD4172
and anti-PD1 therapy. This result is consistent with our *in vitro* experiments, which

suggest that LD4172-induced apoptosis in B16F10 tumors involves TNF/TNFR1
signaling. We have addressed this in lines 238-248, and the relevant figures have been
added to the supplementary data (Figure S6).

Figure S6. Anti-TNF α reversed the synergy between LD4172 and anti-PD1. C57B6/J
mice were subcutaneously inoculated with 3×10^5 B16F10 tumor cells. After seven days
(tumor size ~ 100 mm³), mice were treated every three days with anti-PD1 (100 μ g per
dose, i.p.), daily with LD4172 (20 mg/kg, i.p.), a combination of LD4172 and anti-PD1
(same dose as their individual doses), the combination plus anti-TNF α (200ug every
three days, i.p.), or their corresponding vehicle control (n=5).

Updated paragraph in the manuscript can be found as follows:

Lines 238-248: “LD4172 has also demonstrated significant therapeutic efficacy *in vivo*
by inducing RIPK1 degradation within tumors and exhibiting a synergistic effect on
tumor growth inhibition when combined with anti-PD1 therapy. Although LD4172 plays a
role in inducing ICD in B16F10 tumors, the degradation of RIPK1 alone is insufficient,
as RIPK1 primarily acts as a brake on immunogenic pathways¹⁸. Therefore, additional
ligands are required to fully activate these pathways. Anti-PD1 can supply these
necessary ligands by promoting TNF production, thereby sensitizing cancer cells to cell
death in the absence of RIPK1. This synergistic relationship is further evidenced by the
finding that the combination of LD4172 and anti-PD1 loses its efficacy when anti-TNF- α
is introduced, completely abolishing their combined effect (Figure S6). Moreover, when
used in conjunction with anti-PD1, LD4172 reshapes the tumor immune
microenvironment by enhancing the infiltration of dendritic cells and IFN γ + T cells, as
well as by promoting the secretion of immunostimulatory cytokines, leading to
substantial antitumor effects.”

3. While the proteomic analysis in Fig. 2J indicates a high degree of specificity, an
additional control that could be included is to show that the related kinases RIPK2 and
RIPK3 are not affected by LD4172.

Response: This is an excellent point. Our proteomic data does not rule out the
possibility that RIPK2 and RIPK3 are unaffected by LD4172. To test the selectivity of
LD4172, we used the THP1 cell line, which expresses RIPK1, RIPK2, and RIPK3. After
treating the cells with LD4172 at varying concentrations for 24 hours, we observed
significant degradation of RIPK1 at 16nM, while RIPK2 and RIPK3 levels remained
unchanged, indicating the selectivity of LD4172 for RIPK1. We have addressed this in
lines 122-124, and the relevant figures have been added to the supplementary data
(Figure S2).

Figure S2. Western blot analysis showing the expression levels of RIPK1, RIPK2, and
RIPK3 in THP1 cells following 24-hour treatment with varying concentrations of LD4172.

Updated paragraph in the manuscript can be found as follows:

Lines 122-124: “Additionally, at effective LD4172 concentrations (16 nM–10 μM) that
induce RIPK1 degradation in THP1 cells, we observed no significant changes in the
protein levels of related kinases, such as RIPK2 and RIPK3 (Fig. S2).”

4. In the legend for Figure 3, the early apoptotic cells (AV+, PI-) should be in the upper
left quadrant, not lower right.

Response: Thanks for pointing this out and sorry for our carelessness, the
corresponding correction is highlighted in Figure 3.

5. The key to the graph in Fig S4A for the MC38 experiments appears to be incorrect.
Shouldn't it be PD1 and LD4172 treatment instead of RIPK1 KO?

Response: Thank you very much for catching the mistake. Figure S4A demonstrates the
synergistic effect between LD4172 and Anti-PD1, and the figure legend has been
corrected.

Reviewer #2 (Remarks to the Author): with expertise in cancer, inflammation, RIPK1

The authors have generated a PROTAC targeting RIPK1 for degradation and showed
that it recapitulates the effect of loss of RIPK1 in tumours in vitro and in vivo.
Interestingly, their RIPK1 PROTAC synergises with anti-PD1 therapy in vivo. This study
is generally well designed and controlled. To my knowledge, it is the first RIPK1
PROTAC showing such good synergism in vivo. Given the lethality of RIPK1 mice, this
tool can also be used to study the role of RIPK1 in vivo in adult development, infectious
diseases, etc. There are a few imprecisions and clumsiness throughout the manuscript
that need to be addressed. Please see specific comments.

Specific comments:

In Fig. 2D-E, the authors should acknowledge that RIPK1 is already re-expressed after
a 4-hour washout. This could be important in terms of RIPK1-dependent responses, as
we don't know what level of RIPK1 is required to fully reconstitute its function. For
example, there is a small proportion of RIPK1 that goes to TNF-induced Complex I;
therefore, 10% of RIPK1 (after 4-hour washout) could be enough to provide a normal
TNF response.

Response: Thank you for raising this excellent point. As you suggested, inadequate
RIPK1 degradation can indeed affect the observed phenotype. As demonstrated in
Figure S8, at the late stage of treatment (more than 15 days after tumor inoculation), the
synergy between anti-PD1 and a low dose of LD4172 (20 mg/kg, twice daily) is less
potent compared to a high dose of LD4172 (20 mg/kg, once daily). This indicates that
more thorough and persistent RIPK1 degradation can enhance the synergistic effect.
Additionally, Mannion et al. have shown that intratumoral injection of a RIPK1 degrader
can synergize with anti-PD1 to regress 30% of EO771 breast tumor growth (Figure
adapted from Mannion et al.'s paper). This further suggests that adequate degradation
of RIPK1 in the tumor is more beneficial for promoting cancer immunotherapy, and that
the remaining RIPK1 in tumor might still be sufficient to execute its pro-survival
functions. We have emphasized this point in both **Figure 2E legend and lines 82-84** of
the main text.

Figure S8. Tumor growth curve of mice with B16F10 tumors treated with reduced dosing
 frequency of LD4172. The administration of LD4172 was modified to a reduced dosage
 of 20 mg/kg every other day (n=8). The data are expressed as the mean \pm SEM. *
 $p < 0.1$; ** $p < 0.01$; *** $p < 0.001$; **** $p < 0.0001$. ns, no statistical significance.

[figure redacted]

Figure adapted from Mannion's paper: RIPK1 PROTACs enhance response to immune
 checkpoint blockade

(A) Schematic depicting the treatment regimen of tumor-bearing mice. I.P.,
 intraperitoneal injection; I.T., intratumoral injection.

(B) Tumor growth curves of tumor bearing mice treated as in (A). Thick lines represent
 average tumor growth.

Updated paragraph in the manuscript can be found as follows:

Lines 82-84: "Four hours after the removal of LD4172, RIPK1 starts to resynthesize in
 both cell lines. The resynthesis half-lives are approximately 48 and 24 hours in Jurkat
 and B16F10 cells, respectively (Fig. 2D-E)."

Line 125 'In contrast to situations where RIPK1 is kinase-dead, genetic deletion of
 RIPK1 has been found to trigger apoptosis both in vitro and in vivo 15.'

Ref 15 is only about RIPK1 knock-out mice and not about RIPK1 kinase dead. Plus, ref
 15 is not the only one and the first one to show that loss of RIPK1 induces apoptosis.
 The authors should be fair and cite all RIPK1 kinase dead and RIPK1 knock-out papers
 that support their statement, e.g., Kelliher Immunity 1998; Kaiser PNAS 2014; Rickard
 Cell 2014; Polykratis JI 2014; Berger JI 2014; Newton Science 2014.

Response: Thank you for pointing this out. The recommended papers are at the
forefront of illustrating the kinase-dependent and -independent functions of RIPK1, and
we have cited them in our manuscript.

In Fig S4A the legend says that it's sgNC vs RIPK1 KO, while in the manuscript, the
authors described it as tumours being treated with LD4172. Is it a mistake with the
legend or the wrong graph?

Response: Thanks for pointing this out, it is a mistake with the legend, and we have
fixed it in **Figure S4A legend**.

In Line 181 the authors wrote "Consistent with the in vitro findings, LD4172 also
triggered significant cell death in the tumor (Fig. 182 5A, 2nd column).

The 2nd column is H&E, which does not reflect of cell death but rather absence or
presence of cells. Caspase-3/7 are the cell death markers. Please correct the text
accordingly.

Response: Thanks for pointing this out, we should be strict about description about the
figures, the corrected text is highlighted in main text, **lines 194-197**.

Updated paragraph in the manuscript can be found as follows:

Lines 194-197: "LD4142 treatment significantly disrupted the dense structure of B16F10
tumors, as evidenced by a marked reduction in cellular density observed in H&E
staining (**Fig. 5A, 2nd column**). Importantly, a notable increase in cleaved caspase 3/7
levels was observed in the LD4172-treated tumors, indicating the occurrence of
apoptosis (**Fig. 5A, 3rd and 4th columns**)."

The authors wrote in line 184 'While apoptotic cell death was traditionally considered
non-immunogenic, accumulating experimental data have revealed its potential to drive
immune cell infiltration and anti-cancer immunity 16–19. Supporting the activation of
immunogenic apoptosis, we observed a significant increase in plasma HMGB1 levels
(Fig. 5B) and enhanced exposure of calreticulin on the surface of B16F10 tumor cells
(Fig. 5A, 5th column).'

However, the authors don't have supporting evidence that apoptosis is the mode of cell
death that is immunogenic because it is well established that loss of RIPK1 can also
induce RIPK3-dependent necroptosis (Kaiser PNAS 2014; Rickard Cell 2014; Dillon cell
2014; Berger JI 2014 ; Newton Science 2016), which is also immunogenic. One way to
determine if apoptosis or necroptosis is immunogenic is to analyse MLKL^{-/-} and
casp3/7^{-/-} tumours treated with RIPK1 PROTAC. The authors should rewrite this part.

Response: Thank you for highlighting this important point. In our investigation of the
 kinase-dependent and -independent functions of RIPK1 using LD4172 and T2I, we
 attempted to induce necroptosis in B16F10 cells with a combination of TNF- α , SMAC
 mimetic (LCL161), and the pan-caspase inhibitor z-VAD-FMK. However, even after 72
 250 hours of treatment, the B16F10 cells remained unresponsive to this necroptotic trigger
 (Figure S2), suggesting that RIPK3-dependent necroptosis is not active in these cells.
 This is consistent with reports that B16F10 hardly expresses RIPK3²⁵, which may
 explain their resistance to necroptosis induction. We have emphasized this point in
 lines 140-144, and the relevant figures have been added to the supplementary data
 (Figure S3).

Figure S3: Cell viability of B16F10 cells with indicated treatment (72 h, n = 3).

Updated paragraph in the manuscript can be found as follows:

Lines 140-144: “Although B16F10 cells remained unresponsive to necroptotic triggers
 (TNF α + LCL161 + Z-VAD-FMK, Fig. S3), likely due to the low expression of RIPK3²⁵,
 the combination of TNF α and LD4172 induced significant apoptosis (Fig. 3B-D). This
 was evidenced by the enhanced surface exposure of phosphatidylserine (Fig. 3B) and
 increased levels of cleaved caspase-3/7 and PARP (Fig. 3C-D). Notably, these
 apoptotic effects were reversed with Z-VAD-FMK treatment (Fig. 3B-D).”

The authors wrote in line 196 ‘In addition, combined therapy with LD4172 and anti-PD1
 not only induced extensive TIL infiltration (Fig. 5D-H) but also significantly enhanced
 anti-PD1 positivity in immunologically cold B16F10 tumors, as demonstrated by
 increased infiltration of cytotoxic CD8+ T cells (CD8+IFN- γ +, Fig. 5A, 6th column, and
 5G-H) and decreased infiltration of FOXP3+ T regulatory cells (Fig. 5A, 7th column)
 within the TME.’

This is wrong: LD4172 did not increased TIL infiltration CD8+ T cells induced by anti-
PD1 because there is not statistical difference between anti-PD1 treatment and anti-
PD1+ LD4172 treatment in Fig 5C, G and H. Please correct this part.

Response: Thanks for your careful checks. As you suggested, we were mixing up the
individual effects of LD4172 and anti-PD1 in this part. This has been corrected and
highlighted in lines 209-212.

Updated paragraph in the manuscript can be found as follows:

Lines 209-212: “Additionally, combined therapy with LD4172 and anti-PD1 significantly
enhanced anti-PD1 positivity in immunologically cold B16F10 tumors, as demonstrated
by increased infiltration of cytotoxic CD8+ T cells (CD8+IFN- γ +, Fig. 5A, 6th column,
and 5G-H) and decreased infiltration of FOXP3+ T regulatory cells (Fig. 5A, 7th column)
within the TME.”

The authors wrote in line 235 ‘Unlike Ripk1 knockout mice, which die at 1-3 days of age
due to their widespread roles in multiple tissues and organs 25, homozygous loss-of-
function RIPK1 mutations are well tolerated in humans 26. Patients with complete loss
of RIPK1 protein only showed symptoms confined to the immune system, with primary
immunodeficiency and/or intestinal inflammation 26.’

And Line 240 ‘Although the safety profiles of RIPK1 degraders remain to be tested in
future clinical studies, human genetic data suggest that pharmacological RIPK1
degradation is potentially safe and tolerable, especially with transient intervention in
well-controlled clinical settings.’

And Line 259 ‘Considering the predicted safety profile of RIPK1 degradation based on
human genetics’

By stating that loss of RIPK1 is ‘well tolerated in humans’, ‘only showed symptoms
confined to the immune system’, ‘human genetic data suggest that pharmacological
RIPK1 degradation is potentially safe and tolerable’ and ‘predicted safety profile of
RIPK1 degradation based on human genetics’, the authors are minimising the effect of
loss of RIPK1 in human to promote the use of RIPK1 PROTAC. This is extremely
clumsy because Ref 26 and Li et al PNAS 2019 (doi.org/10.1073/pnas.18135821, which
should be cited alongside Ref 26) showed that RIPK1 deficient patients are affected
with severe immunodeficiency which can lead to the death of the patients at really
young age. The authors should rephrase this part by stating that despite the severe
immunodeficiency cause by permanent loss of RIPK1 in human, acute, chemical and
transient depletion of RIPK1 might be tolerable in humans. Although the safety profiles
of RIPK1 degraders remain to be tested in future clinical studies.

Response: Thank you for your valuable comments. We realize that we were perhaps
too forceful in emphasizing the potential safety of LD4172 and agree that a more critical
and unbiased discussion is necessary. We have revised the discussion accordingly, as
309 per your recommendation, which is reflected in lines 265-270.

Updated paragraph in the manuscript can be found as follows:

Lines 265-270: “Unlike Ripk1 knockout mice, which die within 1-3 days of age due to the
critical role of RIPK1 in multiple tissues and organs³¹, the phenotypes of homozygous
loss-of-function RIPK1 mutations in humans are relatively less severe³². Although
permanent loss of RIPK1 in patients leads to severe immunodeficiency and/or intestinal
inflammation³³, chemical-induced protein degradation of RIPK1, which is acute,
transient, and potentially tissue-specific³⁴, might be more tolerable in humans. However,
the safety profiles of RIPK1 degraders need to be evaluated in future clinical studies.”

Reviewer #3 (Remarks to the Author): with expertise in cancer therapeutics and
degrader design

The authors reported the development of the first-in-class RIPK1 degraders that utilize
the VHL E3 ligase to potently degrade RIPK1 in various cells. One of the lead
compounds, LD4172, is highly specific and only degrades RIPK1 in cells with high D_{max}
and DC₅₀ values in a VHL and proteasome-dependent manner. LD4172 can
synergistically kill B16F10 mouse melanoma cells in combination with TNF α .
Interestingly, administration of B16F10 tumor-bearing mice with LD4172 resulted in
significant reduction in RIPK1 expression in the tumors but not in normal tissues. This
reduction was associated with the potentiation of tumor growth inhibition when it was
combined with anti-PD1 antibody. More importantly, the synergistic effect was greater
with the combination of anti-PD1 antibody and LD4172 than that of anti-PD1 antibody
and the RIPK1 inhibitor. These findings demonstrate that RIPK1 degraders may exert
stronger antitumor activity than RIPK1 inhibitors by degrading RIPK1 to remove both its
kinase and scaffold activity. In addition, this combination can also enhance antitumor
immunity.

Concerns:

1. It is very interesting that only VHL-based PROTACs can effectively degrade RIPK1
but not CRBN- and MDM2-based and adamantane-tagged PROTACs. Can the authors
share some insights why only VHL can degrade RIPK1? Have the authors tried to use
different linker length for other PROTACs to validate the lack of degradation for RIPK1?

Response: Thank you for bringing up this point. In our initial exploration of E3 ligases, we
synthesized two CRBN-based PROTACs with distinct PEG linkers, as detailed in our
manuscript. However, subsequent analysis revealed minimal degradation activity for
RIPK1 with these PROTACs. Following the manuscript submission, we further
investigated this issue and synthesized additional CRBN-based PROTACs with varying
carbon atom linkers, as illustrated in the following Figure. Our western blot results
demonstrated that one of these newly developed CRBN-based PROTAC, T2C10CRBN,
exhibited notable degradation activity for RIPK1 at 100 nM concentration when linker
length reached 10 carbon atoms, albeit with a significant hook effect. However, the
potency of these PROTACs was considerably weaker compared to the VHL-based
PROTAC, LD4172. Currently, LD4172 remains the most potent RIPK1 degrader in our
study. Moving forward, we are dedicated to optimizing CRBN-based PROTACs to
enhance their potency as RIPK1 degraders.

Figure: Chemical structures and potency of new CRBN-based PROTACs.

2. The authors showed that LD4172 can sensitize B16F10 cells to TNF α . Since they
 also showed that LD4172 can synergistically suppress B16F10 tumor growth with Anti-
 PD1 antibody, have the authors tested whether LD4172 can sensitize B16F10 cells to
 cytotoxic T cells?

Response: Thank you for pointing this out. To test whether LD4172 sensitizes B16F10
 cells to cytotoxic T cells, we isolated CD8+ T cells using a mouse CD8a+ T cell isolation
 kit (Miltenyi Biotec) from C57BL/6J mice. These cells were stimulated with 2 \$\mu\$ g/ml plate-
 bound anti-CD3 (BioLegend, clone 145-2C11) and 2 \$\mu\$ g/ml soluble anti-CD28
 (BioLegend, clone 37.51) in RPMI 1640 medium supplemented with 10% FBS, 1% P/S,
 50 nM BME, and 10 ng/mL IL-2 for 2 days. Meanwhile, B16F10 cells were labeled with
 Vybrant™ DiO (ThermoFisher, V22886) and treated with LD4172 (1 \$\mu\$ M) and 1 ng/ml
 IFN \$\gamma\$ for 24 hours. After discarding the medium, we added the activated T cells for 24
 hours at a 1:1 effector to target ratio. The T cells were then removed and killed B16F10
 (DiO+PI \$^+\$ ) cells were analyzed using LSRII. Contrary to our expectations, we observed
 no significant tumor killing, as demonstrated in the figure below. Potential explanations
 include: (1) as the activity of CD8+ T cells are also regulated by other immune cells, it is
 difficult to reconstitute the immune system *in vitro*; (2) Using antigen-specific B16F10
 cells, such as B16F10-OVA, along with OT-1 T cells might be necessary for better
 specificity; or (3) Although we washed away LD4172 before adding T cells to avoid its
 effects on T cells, our washout experiments indicate that B16F10 cells begin to
 resynthesize RIPK1 four hours after LD4172 removal, potentially restoring interactions
 that protect the tumor cells. While this isn't the focus of this manuscript, we plan to
 establish a B16F10-RIPK1-KO-OVA cell line to repeat this experiment in the future.

Figure: Flow cytometric analysis of B16F10 cell killing (DiO⁺PI⁺) after cotreated with
activated T cells for 24 hours.

3. The finding that LD4172 was more effectively in degrading RIPK1 in tumor tissues
than in normal tissues is very interesting. Do the authors explore why these normal
tissues exhibited less RIPK1 degradation after the treatment with LD4172? Is this
related to different uptake of the compounds between tumors and normal tissues?

Response: Thank you for raising this excellent point. Following your recommendation,
we conducted a pharmacokinetic (PK) study by administering a single dose of LD4172
(20 mg/kg) to B16F10 tumor-bearing mice and collected tissue samples at various time
points. Interestingly, although LD4172 uptake in tumors was relatively low, the
compound appeared to be retained within the tumor tissue. Twelve hours post-
administration, the concentration of LD4172 within the tumors remained nearly
unchanged (Figure S9). These findings suggest that the potent degradation of RIPK1
observed in tumors, as compared to normal tissues, may be due to the prolonged
retention of LD4172 within the tumor microenvironment. We have incorporated this
information in lines 279-286, and the relevant figures have been added to the
supplementary data (Figure S9).

Figure S9. LD4172 concentrations in various tissues of B16F10 tumor-bearing
 C57BL/6J mice at different time points following a 20 mg/kg intraperitoneal (i.p.)
 administration (n=4).

Updated paragraph in the manuscript can be found as follows:

Lines 279-286: "Albumin, which constitutes approximately 60% of total plasma protein,
 preferentially accumulates in tumors due to the high demand for amino acids and
 energy in these tissues^{35,36}. Given that 98.6% of LD4172 is bound to plasma proteins
 (**Table 1**), it is plausible that LD4172 may be "piggybacking" on albumin accumulation in
 tumors, thereby achieving tumor-selective RIPK1 degradation. Supporting this
 hypothesis, following a single 20 mg/kg dose of LD4172 in C57BL/6J mice bearing
 B16F10 tumors, LD4172 was found to persist in the tumor for an extended period
 compared to other tissues (**Figure S9**). This prolonged retention in the tumor could
 further mitigate potential toxicity concerns related to RIPK1 degradation in normal
 tissues."

4. RIPK1 KO has a profound effect in mice as they die at 1-3 days of age while patients
 with homozygous loss-of-function RIPK1 are alive and only exhibit some less server
 abnormalities such as primary immunodeficiency and/or intestinal inflammation.
 However, the mice treated with LD4172 did not show any side effects in mice. This may
 be attributable to the inability of LD4172 to effectively degrade RIPK1 in normal tissues.
 This finding may have important implications in developing RIPK1 targeted therapy
 because RIPK1 might be more toxic to normal tissues while RIPK1 PROTACs might be
 more selective to tumors. As such, RIPK1 PROTACs may have a better opportunity to
 be developed as anticancer drugs.

Response: Thank you for pointing this out. We agree with the reviewer that transient
RIPK1 degradation induced by LD4172 may have very different pharmacology from
persistent RIPK knockout in cells. In addition, the tissue selectivity of RIPK1
degradation included by LD4172 may be another intriguing feature to further develop
these degraders as therapeutics.

5. The finding that depletion of CD8 T cells with an antibody abrogated the synergistical
antitumor activity of LD4172 and anti-PD1 antibody suggests that the direct antitumor
activity of RIPK1 PROTACs is less important than the induction of CD8 T cell-mediated
antitumor immunity in the B16F10 tumor model.

Response: Thank you for pointing this out. In all the cancer cell lines we tested, we did
not observe toxicity or inhibition of proliferation induced by LD4172 alone. However,
degradation of RIPK1 rewires the cancer cell death pathways and also boost the
anticancer immunity through interacting with immune cells. We view this RIPK1
degrader more like a catalyst than an executioner.

Reviewer #4 (Remarks to the Author): with expertise in cancer therapeutics and
degrader design

I support publication of this work after addressing comments below.

• What are the noteworthy results?

This is a nice study describing the discovery and characterization of the first PROTAC-
based RIPK1 degrader. The study appears to be done with excellent quality and the
conclusions are well supported by the provided data.

Key results:

- 1. RIPK1 can be degraded using a suitably designed/optimized PROTAC degrader
- 2. RIPK1 degrader has fundamentally different pharmacology relative to RIPK1 inhibitor
- 3. Degradation of RIPK1 by LD4172 triggered immunogenic cell death (ICD)
- 4. Degradation of RIPK1 resulted in an increase in tumor-infiltrating lymphocytes and
sensitized tumors to anti-PD1 therapy

The efficacy in the mouse models is limited (this would be called progressive disease in
patients) but does show the mechanism appears to be active in a murine model.

It would be nice to see:

- 1. PK data for LD4172 to enable understanding of how much drug exposure there was
following the dosing regime used.

Response: Thank you for highlighting this important point. In the initial phase of the
LD4172 PK study, we aimed to use the lowest effective dose and did not include the 20
mg/kg dosing regimen. Based on your suggestion, we repeated the PK study with a 20
mg/kg (i.p.) dose of LD4172, measuring its concentration in various tissues at different
time points (Figure S8). Consistent with our previous findings, LD4172 was rapidly
metabolized and cleared from the blood. However, in B16F10 tumors, LD4172 was
found to persist in the tumor for an extended period compared to other tissues. This
prolonged retention in the tumor could further mitigate potential toxicity concerns related
to RIPK1 degradation in normal tissues. The relevant figure has been added to the
supplementary data (Figure S9).

**Figure S9.** LD4172 concentrations in various tissues of B16F10 tumor-bearing
 C57BL/6J mice at different time points following a 20 mg/kg intraperitoneal (i.p.)
 administration (n=4).

2. To show evidence of target degradation in vivo (blood and tumor tissue).

Response: Thank you for bringing up this crucial point, which has been a key focus in
 the development of LD4172. We conducted a pharmacodynamic (PD) experiment in
 C57BL/6J mice bearing B16F10 tumors with intraperitoneally delivery (i.p.) of indicated
 dosages (Figure 4B). Among the various tissues tested, we observed that potent RIPK1
 degradation occurred primarily in the tumors. To further understand this, we performed a
 pharmacokinetic (PK) study to measure LD4172 concentrations across different tissues.
 Interestingly, while B16F10 tumors did not exhibit the highest uptake of LD4172, the
 clearance of the compound in the tumor was slower, allowing for a longer duration of
 RIPK1 degradation. Additionally, we explored intratumoral delivery (i.t.) of LD4172 in
 C57BL/6J mice, which resulted in more pronounced RIPK1 degradation compared to
 intraperitoneal administration (Figure S7). Moving forward, we plan to optimize LD4172
 to improve its penetration and efficacy in tumor tissues.

C57BL/6J Mice-LD4172 (2 doses a day, 3 consecutive days)

Figure 4B. Representative immunoblots of RIPK1 in different tissues of C57BL/6J mice
treated with LD4172 (n=3-4).

Figure S7. Pharmacodynamic (PD) properties of LD4172 with intratumoral
administration (i.t.). Representative immunoblots of RIPK1 expression in various tissues
of C57BL/6J mice treated with LD4172 are shown. Mice with syngeneic B16F10 tumors
received intratumoral injections of LD4172 (10mg/kg) twice daily for three days. Upon
sacrifice, tissues were collected, and the levels of RIPK1 were quantified through
Western blotting.

• Will the work be of significance to the field and related fields? How does it compare to
the established literature? If the work is not original, please provide relevant references.

Yes demonstrates another use of small molecule degraders to achieve novel
pharmacology.

• Does the work support the conclusions and claims, or is additional evidence needed?

The work is well supported by the data provided.

• Are there any flaws in the data analysis, interpretation and conclusions? Do these
prohibit publication or require revision?

None that I could find.

• Is the methodology sound? Does the work meet the expected standards in your field?

• Is there enough detail provided in the methods for the work to be reproduced?

Yes experimental sections appear to be sound, nice characterization data is provided for
the compounds.
